# MiR-223-3p in Cancer Development and Cancer Drug Resistance: Same Coin, Different Faces

**DOI:** 10.3390/ijms25158191

**Published:** 2024-07-26

**Authors:** Davide Barbagallo, Donatella Ponti, Barbara Bassani, Antonino Bruno, Laura Pulze, Shreya A. Akkihal, Jonahunnatha N. George-William, Rohit Gundamaraju, Paola Campomenosi

**Affiliations:** 1Department of Biomedical and Biotechnological Sciences, Section of Biology and Genetics “Giovanni Sichel”, University of Catania, Via Santa Sofia 89, 95123 Catania, Italy; 2Interdisciplinary Research Centre on the Diagnosis and Therapy of Brain Tumors, University of Catania, Via Santa Sofia 78, 95123 Catania, Italy; 3Department of Medical-Surgical Sciences and Biotechnologies, University of Rome Sapienza, Corso della Repubblica 79, 04100 Latina, Italy; donatella.ponti@uniroma1.it; 4Laboratory of Innate Immunity, Unit of Molecular Pathology, Biochemistry, and Immunology, Istituto di Ricovero e Cura a Carattere Scientifico (IRCCS) MultiMedica, Via Fantoli 16/15, 20138 Milano, Italy; barbara.bassani@multimedica.it (B.B.); antonino.bruno@uninsubria.it (A.B.); 5Department of Biotechnology and Life Sciences, DBSV, University of Insubria, Via J.H. Dunant 3, 21100 Varese, Italy; laura.pulze@uninsubria.it; 6Independent Researcher, 35004 SE Swenson St, Snoqualmie, WA 98065, USA; sankolekar@gmail.com; 7Department of Medical Biotechnology and Translational Medicine, University of Milan, Via Fratelli Cervi, 93, 20054 Segrate, Italy; jonahnesson@gmail.com; 8Department of Laboratory Medicine, University of California San Francisco, 513 Parnassus Avenue, San Francisco, CA 94143, USA; rohit.gundamaraju@utas.edu.au; 9ER Stress and Mucosal Immunology Team, School of Health Sciences, University of Tasmania, Launceston, TAS 7248, Australia

**Keywords:** biomarker, cancer, cancer immunity, chemoresistance, diagnosis, inflammation, microRNA, miR-223-3p, prognosis, radioresistance

## Abstract

MicroRNAs (miRNAs) are mighty post-transcriptional regulators in cell physiology and pathophysiology. In this review, we focus on the role of miR-223-3p (henceforth miR-223) in various cancer types. MiR-223 has established roles in hematopoiesis, inflammation, and most cancers, where it can act as either an oncogenic or oncosuppressive miRNA, depending on specific molecular landscapes. MiR-223 has also been linked to either the sensitivity or resistance of cancer cells to treatments in a context-dependent way. Through this detailed review, we highlight that for some cancers (i.e., breast, non-small cell lung carcinoma, and glioblastoma), the oncosuppressive role of miR-223 is consistently reported in the literature, while for others (i.e., colorectal, ovarian, and pancreatic cancers, and acute lymphocytic leukemia), an oncogenic role prevails. In prostate cancer and other hematological malignancies, although an oncosuppressive role is frequently described, there is less of a consensus. Intriguingly, *NLRP3* and *FBXW7* are consistently identified as miR-223 targets when the miRNA acts as an oncosuppressor or an oncogene, respectively, in different cancers. Our review also describes that miR-223 was increased in biological fluids or their extracellular vesicles in most of the cancers analyzed, as compared to healthy or lower-risk conditions, confirming the potential application of this miRNA as a diagnostic and prognostic biomarker in the clinic.

## 1. Introduction

MicroRNAs (miRNAs) are 20–24-nucleotide-long non-coding RNAs that fine-tune gene expression at the post-transcriptional level [1]. MiRNA genes can: (i) have their own promoter and regulatory sequences; (ii) be contained within other genes, often in introns; or (iii) be organized in clusters, whose members are co-regulated [1]. MiRNA biogenesis is complex, involving multiple enzymes and proteins [2,3]. Comprehensive reviews are available on miRNA genes regarding their general organization, expression, biogenesis, and physiological functions [4].

A single miRNA may regulate the expression of hundreds of protein-coding genes [5]. MiRNAs pleiotropically regulate various biological functions, such as differentiation, development, proliferation, apoptosis, angiogenesis, etc. [6,7,8]. Recently, scientists have increasingly focused on miRNAs, unraveling their multifaceted physiological roles, and have noted their dysregulation in pathological conditions. Armed with newly discovered tools, scientists are conducting in-depth miRNA studies—including miRNA cross-species functional annotations and more powerful predictions of miRNA targets—to ascertain their pathophysiological roles [8]. Analyzing miRNA binding sites provides valuable insights into the regulatory relationship between miRNAs and their targets [9,10]. Using these methods, as many as 1917 mature miRNA sequences have been discovered so far (miRBase, accessed on 1 September 2023) [10]; together, they modulate > 60% of the protein-coding genes in humans [11,12].

Recently, it has become evident that other non-coding RNAs, such as long-noncoding RNAs (lncRNAs) and circular RNAs (circRNAs), interact with miRNAs to control gene expression through multiple layers of complexity [12,13]. A multitude of circRNAs act as miRNA sponges and decoys for RNA-binding proteins, actively regulating onco-suppressors or oncogenes through complex competitive endogenous RNA (ceRNA) networks [13,14,15,16,17,18]. Compared to linear RNA transcripts, circRNAs are more resistant to RNase R activity, consequently having longer half-lives within cells [15,16,17,19].

Lately, the crucial role of miRNAs in cancer etiopathogenesis and progression has been becoming more evident. Various endogenous and exogenous factors (stress, environmental factors, epigenetic alteration, genes mutation, drugs, etc.) cause miRNA dysregulation in cancer cells [20,21,22]. Advances in constructing miRNA profile databases, isolating RNA from difficult samples (such as biological fluids or extracellular vesicles (EVs)), and implementing new high-throughput gene expression technologies (such as DNA microarrays and RNA sequencing) have further fostered the research on the involvement of miRNAs in cancer [22,23]. These studies, which reveal that miRNA expression is modulated depending on different types and subtypes of cancer, allow us to rebuild molecular mechanisms underlying their dysregulation. Further, it is becoming evident that the dual oncogenic and oncosuppressive role of miRNAs in different cancer types is (at least in part) due to the tissue-specific gene expression profiles [24,25,26,27].

Significant improvements in overall survival rate, reduction in tumor growth, and inhibition of metastasization in cancer patients can be achieved in two ways: (i) by silencing oncogenic miRNAs, which consequently will induce the expression of tumor suppressor genes; and (ii) by overexpressing oncosuppressive miRNAs that reduce the expression of oncogenes [28]. MiRNAs are also responsible for the development of chemoresistance in various cancer types [29]. Interestingly, modulating miRNA expression in patients undergoing relapse can halt cancer progression and sensitize cancer cells to specific drugs [30]. Further, because they are: (i) easily identifiable (by techniques such as sequencing, PCR, and in situ hybridization (ISH)); (ii) particularly resistant to degradation; (iii) evolutionarily conserved; and (iv) powerful regulators of gene expression, miRNAs can be handy clinical tools in treating cancers that still lack diagnostic, therapeutic, and prognostic biomarkers [31,32].

Here, we will briefly describe miR-223-3p’s (MIMAT0000280) involvement in immune response and inflammation (Section 2) and describe in detail the role of this peculiar miRNA in tumorigenic and anti-cancer therapy response processes (especially in the response to treatments targeting solid and hematological malignancies) (Section 3).

## 2. Physiological and Pathophysiological Roles of MiR-223-3p: Inflammation and Immune Cell Differentiation

MiR-223-3p (henceforth miR-223), together with miR-223-5p, is the mature miRNA product of the *MIR223* gene, which maps on chromosome Xq12 (https://www.ncbi.nlm.nih.gov/gene/407008, accessed on 10 July 2024) [33]. The first-described and best-recognized role of miR-223 is the modulation of hematopoietic lineage differentiation [34]. It is also involved in regulating human embryonic stem cell (hESC) [35] and osteoclast differentiation [36].

MiR-223 exerts an important role in the differentiation and activation of cells of the immune system. The latter, together with inflammation, is crucially involved in cancer development. Although the relevance of miR-223 in hematopoietic cell development and its contribution to innate and adaptive immune responses have been comprehensively reviewed before [37,38], we summarize the most important features here.

MiR-223 modulates innate immunity, crucially regulating the maturation, differentiation, and polarization states of several immune cells, including neutrophils, monocytes, and dendritic cells [37]. MiR-223 is absent in B and T lymphocytes, while it is highly expressed in CD34^neg^ hematopoietic cells within the bone marrow and, in particular, in mature granulocytes, thus suggesting an association of miR-223 expression with this differentiation lineage [39]. Indeed, Vian and colleagues demonstrated that miR-223 overexpression is associated with increased granulopoiesis and with the concomitant inhibition of erythropoiesis [40]. These results are partially contradictory to the data provided by Johnnidis and colleagues, who found that miR-223 negatively regulates granulopoiesis [34]. In their study, hemizygous *Mir223^−^*/Y mice showed a cell-autonomous increase in granulocyte progenitors, which caused an increase in the number of circulating and bone marrow granulocytes. They showed that miR-223 targets the transcription factor myocyte enhancer factor 2C (Mef2c). Genetic ablation of *Mef2c* rescued the neutrophilic phenotype in miR-223 null mice [34]. Moreover, granulocytes lacking miR-223 were hyperactive—they were “hypermature, hypersensitive to activating stimuli, and displayed increased fungicidal activity”—which caused miR-223 mutant mice to suffer from spontaneous lung inflammation and excessive tissue destruction in response to endotoxin challenge [34].

During myelopoiesis in humans, *MIR223* transcription is regulated through two regulatory regions: (i) distal and (ii) proximal. The distal regulatory region (approximately 3400 bp from the 5′ end of the pre-miR-223) contains cis-regulatory elements for the transcription factors Spi-1 proto-oncogene (PU.1) and CCAAT enhancer binding protein beta (C/EBPβ). The proximal regulatory region (1000 bp upstream of the pre-miR-223) is regulated by the competitive binding of CCAAT-box-binding transcription factors NFIA and CCAAT/enhancer-binding protein α (C/EBPα)—both vying for the same CAAT element [39]. During monocytic differentiation, *MIR223* transcription is increased by the recruitment of C/EBPβ and PU.1 to the distal regulatory region, while the recruitment of C/EBPα to the proximal regulatory region drives the strong induction of miR-223, as seen during granulopoiesis. On the contrary, during erythroid differentiation, TAL bHLH transcription factor 1, erythroid differentiation factor (TAL-1), LIM-domain-only 2 (LMO2), and GATA-binding protein 1 (GATA-1) repress *MIR223* transcription by binding to their respective sites on the proximal regulatory region [39]. Notably, one of the targets of miR-223 is NFIA, suggesting that increased miR-223 levels decrease NFIA expression, creating a positive feedback loop.

Consistent with its involvement in myeloid differentiation, in promyelocytic leukemia cells, miR-223 levels modulate the expression of the monocytic and granulocytic differentiation marker CD11b; enhanced miR-223 levels induce CD11b expression, while decreased miR-223 levels reduce CD11b expression [39].

MiR-223 affects immune cell functions by targeting the mRNAs of the general sensors of cellular stress (such as NLRP3: the nucleotide-binding oligomerization domain (NOD)-like receptor family pyrin domain-containing 3 protein) in murine neutrophils and macrophages [41]. Feng and colleagues showed that miR-223 crucially inhibits the activity of the NLRP3 inflammasome to alleviate the acute lung injury/acute respiratory distress syndrome (ALI/ARDS) induced by mitochondrial damage-associated molecular patterns (DAMPs) [42]. Moreover, miR-223 inhibits the generation of neutrophil extracellular traps (NETs) by granulocytes. Indeed, high miR-223 levels in neutrophils restrain mitochondrial ROS production by blocking Ca^2+^ influx, in turn suppressing interleukin (IL)-18-mediated NETs formation. Additionally, increased levels of neutrophil-derived exosomal miR-223 suppresses NLRP3 inflammasome formation by dampening IL-18 production in macrophages [43]. Accordingly, Ye and colleagues showed that, in an acute liver failure (ALF) mouse model, miR-223 deficiency increases in vivo neutrophil elastase (NE) production, thereby enhancing NETs formation [44]. The livers of ALF mice showed NET formation, which was markedly increased in miR-223 KO mice. ALF-related hepatocellular damage and mortality (which are induced by an increased neutrophil infiltration) were significantly higher in miR-223 KO mice compared to wild-type controls [44].

MiR-223 critically regulates macrophage polarization. MiR-223 downregulation promotes the M1 (pro-inflammatory) phenotype while restraining the polarization towards the M2 (anti-inflammatory) phenotype. Indeed, in bone marrow macrophages isolated from *Mir223*^−^/Y mice, the lipopolysaccharide (LPS) or poly(I:C) mediated stimulation of Toll-like receptor 4 (TLR4) activates signal transducers and the activator of transcription 3 (STAT3), thus supporting the production of pro-inflammatory cytokines (including IL-6 and IL-1β, but not TNF-α) and M1 polarization [45,46]. Li and colleagues showed that miR-223 is downregulated during macrophage differentiation, priming them to the activation of the NF-κB non-canonical pathway in case of an inflammatory response due to the upregulation of its target, CHUK (encoding IKKα, a component of inhibitor of nuclear factor kappa B kinase complex) [47]. Conversely, miR-223 indirectly downregulates TLR4 protein levels and receptor signaling, limiting the development of atherosclerotic plaques in *Apoe*^−/−^ mouse models [48]. Furthermore, the transcription factor PBX/knotted 1 homeobox 1 (Pknox1), a direct target of miR-223, promotes M1 polarization of murine macrophages. Finally, following IL-4 stimulation, peroxisome proliferator-activated receptor gamma (PPARγ) induces *Mir223* transcription; miR-223, in turn, targets RAS p21 protein activator 1 (Rasa1) and nuclear factor of activated T cells 5 (Nfat5), which drives macrophage polarization towards the M2 state. MiR-223 deletion in an obese murine model prevented induction of the alternative macrophage polarization by PPARγ [49].

Notably, miR-223 is one of the most abundant miRNAs in EVs [38,45,50]. EVs deliver miR-223 to target cells, affecting their behavior. This may represent an important crosstalk mechanism between stromal and cancer cells, such as those occurring between macrophages and cancer cells in breast and ovarian cancers [51,52].

We will now proceed to review miR-223 roles—mediated by different and often contrasting molecular mechanisms—in specific cancer types and subtypes, as well as in the anti-cancer treatment response.

## 3. Role of miR-223 in Specific Types of Cancers

The role of miR-223 in cancer is generally attributed to targeting specific transcripts. Figure 1 and Figure 2 facilitate the comparison of the targets mediating the oncosuppressive or oncogenic roles of miR-223 across the different types of cancer.

### 3.1. MiR-223 in Colorectal Cancer (CRC)

Colorectal cancer (CRC) is the third and the second most common cancer in men and women worldwide, respectively [53,54]. While its incidence rates correlate with the level of socioeconomic development [55], diet, obesity, limited physical activity, smoking habit, and genetics are the main risk factors. Genetics also play an important role in hereditary forms of CRC [56].

CRCs can arise from normal epithelium, through benign lesions, and via several pathways. The best-known CRC genesis pathway is the adenoma to adenocarcinoma pathway, first described by Fearon and Vogelstein in 1990 [57]. CRC can exhibit chromosomal instability (CIN), microsatellite instability (MSI), or CpG island methylator phenotype (CIMP) [58,59]. Like in other types of cancer, miRNAs are involved in the development and progression of CRC [60].

#### 3.1.1. MiR-223 as an Onco-Suppressor in CRC

Evidence for oncosuppression by miR-223 in CRC is very scarce, as outlined by the studies indicated in Table 1. Wu and colleagues found that, in HCT116 cells (a human CRC cell line), ectopically expressed miR-223 downregulates forkhead box O1 (FOXO1) and modulates its nuclear/cytoplasmic distribution ratio, ultimately altering the levels of cyclin D1/p21/p27. Further, miR-223 overexpression inhibits HCT116 cell proliferation [61]. The oncogenic lncRNA ROR is associated with various human cancers, including CRC. In CRC, the upregulated lncRNA ROR sponges miR-223 and inhibits NF2 onco-suppressor protein expression, thereby increasing cancer cell proliferation and invasion [62].

However, notably, evidencing its inflammation-regulatory roles, miR-223 is increased in both IBD patients and murine colitis models [63]. A well-known azoxymethane/dextran sulfate sodium-induced mouse model of colitis-associated cancer was used to analyze miRNAs and mRNAs modulated during inflammation-induced tumor development. MiR-223 was one of the most upregulated miRNAs in this model, and it was found to inhibit AKT serine/threonine kinase 1 (Akt) phosphorylation and insulin-like growth factor 1 receptor (IGF-1R) expression [64].

#### 3.1.2. MiR-223 as an Onco-miRNA in CRC

Several studies show that miR-223 is increased in CRC tissue compared to normal tissue [65,66,67], suggesting that it is an onco-miRNA, as indicated in Table 1. However, the possible differential role of miR-223 in the different types of CRC (i.e., with chromosomal instability or microsatellite instability) has yet to be investigated.

While analyzing 55 paired CRC and non-neoplastic mucosa samples using a 24-miRNA panel (which were identified as being altered in cancers through a literature search), Earle and colleagues found that miR-223 was upregulated in CRC tissues compared to non-neoplastic tissues, particularly in patients with Lynch syndrome (hereditary non-polyposis colorectal cancer, HNPCC) [66]. Using a multi-step approach that combined miRNA-microarray, bioinformatics, and Pearson’s correlation analysis, Fu and colleagues confirmed that miR-223 is upregulated in CRC tissues compared to normal tissues [67]. In another microarray screening of a large cohort of CRC samples, Zhang and colleagues found higher expression of miR-223 in CRC tissue than in control tissue. Moreover, they found that higher levels of miR-223 positively correlated with the clinical stage and that the inhibition of miR-223 decreased cell proliferation, migration, and invasion in Colo320 and LoVo CRC cell lines [65]. Chai and colleagues found increased levels of miR-223 in colon cancer compared to normal colon tissues [68]. They showed that miR-223 targets PR/SET domain 1 (PRDM1) (also known as “B lymphocyte-induced maturation protein” (Blimp-1)) in RKO and HCT116 cell lines, stimulating proliferation, migration, and invasion while inhibiting apoptosis; PRDM1 is a zinc finger transcription factor regulating downstream genes such as the *β-interferon* gene [68].

MiR-223 interferes with cell death mechanisms in CRC cells and increases cell proliferation by downregulating BCL2-like 11 (BCL2L11/BIM) expression through forkhead box O3 (FOXO3a) [69]. In a study conducted in SW620 cells, miR-223 negatively regulated FoxO3a/BIM signaling, thereby promoting cell proliferation [70]. MiR-223 directly targets the *F-box and WD repeat domain-containing 7* (*FBXW7*) transcript, which is the substrate recognition subunit of SCF (SKP1/CUL1/F-box protein) E3 ligase complex, thus repressing apoptosis and promoting proliferation via the Notch and Akt/mTOR pathways [71,72]. MiR-223 expression is also negatively correlated with the expression of the tumor suppressor Ras homolog family member B (RhoB). Indeed, miR-223 inhibition elevates RhoB expression, thus affecting the proliferation of LS174, T84, and HT29 colorectal adenocarcinoma cell lines [73]. RAS-GTPase-activating proteins (RASGAPs) are RAS signaling terminators associated with tumor progression/tumorigenicity. In an attempt to characterize the involvement of RASA1 in CRC, Sun and colleagues found that its mRNA is a target of miR-223, whose expression is high and regulated by the C/EBP-b transcription factor in CRC [74]. Treating Caco-2 cells with anti-miR-223 increased RASA1 expression; further, injecting HT29 cells ectopically expressing either pre-miR-223 or anti-miR-223, correspondingly, increased or decreased the tumor growth in nude mice [74]. MiR-223, along with miR-106a-5p are hub miRNAs for post-transcriptional control of *solute carrier family 4 member 4* (*SLC4A4*), a gene product with pleiotropic oncosuppressive functions whose expression is downregulated in CRC; increased expression of both miRNAs was described in CRC samples [75]. LncRNA DRAIC may sponge miR-223 in CRC cells, inhibiting tumor cell proliferation [76]. Also, circLRCH3 was found to sponge miR-223 in CRC. Indeed, circLRCH3 levels are inversely correlated with miR-223 in CRC, and miR-223 silencing decreases the growth and proliferation of cancer cells [77].

#### 3.1.3. Role of miR-223 in Drug Resistance in CRC

The MiR-223/FBXW7 axis regulates doxorubicin resistance in CRC cells. MiR-223 suppression increases FBXW7 expression, which chemo-sensitizes CRC cells to doxorubicin. Additionally, the miR-223/FBXW7 axis orchestrates the epithelial-to-mesenchymal transition (EMT) process in colon cancer; this knowledge can be utilized in tailoring personalized anti-cancer treatments [78]. Cisplatin-resistant gastric cancer cells also show high miR-223 and low FBXW7 expressions [78]. A list of relevant studies on the topic is illustrated in Table 1.

#### 3.1.4. MiR-223 as a Biomarker in CRC

MiR-223 could be an early diagnostic and prognostic biomarker of CRC progression in clinical setting (Table 2). Li and colleagues showed higher miR-223 expression in CRC tissues compared to adjacent non-cancerous tissues. Its expression increases with TNM staging, and distant and lymph node metastases, and its expression correlates with poor overall survival [79]. MiR-223 is a highly sensitive biomarker of early-stage CRC; Mahmoud and colleagues found higher miR-223 serum levels in patients with CRC compared to healthy controls. They also showed that higher miR-223 and miR-182 serum levels are associated with an increased risk of CRC progression [80]. Plasma miR-223 levels are elevated in patients with CRC and correlate with advanced TNM staging and lymph node metastases. Moreover, plasma miR-223 levels are significantly reduced after surgical removal of tumors in patients with CRC [81].

Profiling the miRNAome of serum samples from patients affected by colorectal adenoma or adenocarcinoma and healthy controls (using next generation sequencing) revealed miR-223 as a potential diagnostic biomarker of colorectal adenocarcinoma. Its relevance as diagnostic biomarker was confirmed in a qRT-PCR validation study, which revealed a gradual increase in miR-223 levels from stage I to stage IV [82]. Another study confirmed that high levels of serum-derived miR-223 can be used as a diagnostic biomarker of CRC [83]. Combined changes in miR-223 and miR-92a levels in plasma and stool (obtained from fecal occult blood test) were shown to be a promising diagnostic biomarker for CRC. The two miRNAs together yielded 96.8% sensitivity and 75% specificity for CRC detection [84]. Microarray analysis revealed higher miR-223 expression in stool samples of ethnically Asian patients with CRC [85]. Conversely, Chang and colleagues found decreased miR-223 levels in the stools from patients with CRC compared to those from normal volunteers [84]. Higher miR-223 and miR-182 levels in the serum of CRC patients are associated with an increased risk of CRC progression [80]. In a study evaluating serum miRNA biomarkers of early-stage CRC in a Japanese cohort using a human miRNA oligo chip comprising 2555 miRNAs, followed by three rounds of validation on different small cohorts, miR-223 was found to exhibit the highest diagnostic accuracy (AUC = 1) in one validation set, but a fair accuracy in the following validations (AUC = 0.632 and 0.680), suggesting that some differences in the cohorts affected the accuracy [86].

### 3.2. MiR-223 in Lung Cancer

Although lung cancer ranks among the top five tumors in terms of incidence, it ranks first for mortality [54,87]. It is a heterogeneous disease comprising small-cell lung cancer (SCLC), representing 15% of lung cancers, and non-small-cell lung cancer (NSCLC), accounting for 85% of lung cancers and including adenocarcinomas (ADC, 55% of all NSCLCs) and squamous cell carcinomas (SCC, 45% of all NSCLCs). Among the risk factors, smoking plays a major role, although lung adenocarcinoma does also occur in non-smokers. Besides playing a direct mutagenic role, several components of cigarette smoke induce a state of inflammation that promotes tumorigenesis and alters the gene expression program in cells [88,89]. NSCLC is characterized by a high mutational burden [90], which makes it amenable to therapy with immune checkpoint inhibitors. Although overall mortality is still high, earlier detection by low-dose computed tomography and advances in treatment are contributing to extended survival [54]. Further improvement in diagnosis may be obtained by identifying minimally invasive biomarkers, such as miRNAs. Among these, circulating miR-223 is a very promising candidate (see below).

As described in Section 2, miR-223 is associated with inflammation. Given that smoking induces a state of chronic inflammation and that miR-223 is crucial to this process, its modulation is expected to cooperate in inflammatory-driven carcinogenesis. Indeed, this miRNA is downregulated in bronchial brushings of smokers as compared to those of former smokers or never-smokers, and also in bronchoscopy samples from patients with lung cancer as compared to those from patients with benign lesions [89,91]. Izzotti and colleagues, using a rat model of environmental exposure to cigarette smoke, showed that miR-223 is among the most downregulated miRNAs in these animals’ lung tissues [92]. Treatment of A549 cells with benzo[a]pyrene decreases miR-223 and increases NLRP3 levels, promoting cell pyroptosis as a regulated cell death mechanism [93]. However, the molecular mechanisms by which miR-223 contributes to tumor-promoting inflammation and to lung tumorigenesis remain unclarified, especially in the case of SCLC. Here, we review the state of the art concerning this knowledge, focusing on NSCLC.

#### 3.2.1. MiR-223 as an Onco-Suppressor in NSCLC

MiR-223 was identified as an onco-suppressor (Table 1) during an integrative analysis of miRNA and mRNA sequencing data from matched lung ADC and normal samples; further, ectopic expression of miR-223 in NSCLC cell lines A549 and NCI-H460 reduced colony formation [94]. The onco-suppressive role of miR-223 was also demonstrated in lung SCC by Luo and colleagues, who found that miR-223 expression was downregulated in SCC tissues compared to adjacent normal lung tissues. Downregulation was even more marked in SCC tumor tissues that successfully formed xenografts in mice compared to those that failed. Successful xenografting was associated with *TP53* mutations. The authors also showed that mutant p53 represses miR-223 expression by directly binding to the *MIR223* promoter. In turn, TP53 3′UTR is targeted by miR-223, creating a feedback loop affecting lung SCC proliferation and metastasis. Moreover, Luo and colleagues demonstrated that miR-223 inhibits cell survival, proliferation, and migration in the SCC cell lines SK-MES-1 and NCI-H520 [95].

MiR-223 suppresses NSCLC by targeting different mRNAs. One of its best-characterized targets is NLRP3, a receptor involved in inflammasome and innate immune system activation. MiR-223 suppresses NLRP3-induced invasion and migration of NSCLC cell lines A549 and H520 [96]. Moreover, SLCO4A1-AS1 lncRNA (which is upregulated in NSCLC tissues compared to normal lung tissue) sponges miR-223, leading to increased levels and activation of CHUK and the NF-κB pathway, which, again, are linked to inflammation [97]. Another miR-223 target is pleiotrophin (PTN), a secreted heparin-binding growth factor with roles in cell growth, survival, stemness, cell migration, and angiogenesis. Peng and colleagues showed that in lung SCC, lncRNA PITPNA-AS1 inhibits the PTN-targeting mediated onco-suppressive activity of miR-223 [98].

Mateu-Jiménez and colleagues found that miR-223 is downregulated in murine LP07 cell line lung tumors formed in isogenic BALB/c *Parp1*^−/−^ or *Parp2*^−/−^ mice compared to those formed in wild-type animals [99]. Furthermore, based on previous data showing that miR-223 regulates PARP-1 expression in a model of pulmonary hypertension [100], they proposed that a negative feedback loop exists between PARP proteins and miR-223 in lung adenocarcinoma [99].

#### 3.2.2. MiR-223 as an Onco-miRNA in NSCLC

Reflecting the heterogeneous behavior of miR-223 in NSCLC, its oncogenic properties have also been described (Table 1). Li and colleagues found that inhibiting miR-223 decreases NSCLC cell viability, migration, and invasion, and moreover, it induces apoptosis by preventing its targeting of RhoB [101]. MiR-223, in platelet-derived microvesicles isolated from the blood of patients with NSCLC, modulates A549 cell adhesion and invasion by targeting erythrocyte membrane protein band 4.1-like 3 (EPB41L3), a member of the band 4.1 family of cytoskeletal proteins [102].

Liu and colleagues found that miR-223 promotes proliferation, migration, and invasion of A549 and SPC-A1 lung cancer cells by targeting Transforming Growth Factor Beta Receptor 3 (TGFBR3) [103]. LncRNA ADAMTS9-AS2, which is downregulated in NSCLC, acts as an onco-suppressor by sponging miR-223 to prevent TGFBR3 targeting [103].

#### 3.2.3. Role of miR-223 in Drug Resistance of NSCLC Cells

The role of miR-223 in drug resistance (Table 1) observed in patients with NSCLC is controversial. MiR-223 confers resistance to standard chemotherapeutics, such as cisplatin [104] and doxorubicin [105], by targeting FBXW7 and increasing NSCLC autophagy and EMT [104,105]. Targeting of FBXW7 by miR-223 may also mediate the onset of erlotinib resistance in the NSCLC cell line HCC827. MiR-223 upregulation is attributed to the activation of Akt and Notch pathways, which causes an increase in cancer stemness and related phenotypes, such as EMT, migration, and colony formation [106].

Conversely, Kuo and colleagues found that patients with NSCLC who have low miR-223 levels had higher histamine N-methyltransferase (HNMT) expression in cancer tissue (compared to normal tissue) and a worse prognosis than those with low HNMT and higher miR-223 expression in their cancer tissue. HNMT upregulation affects HER2 expression and increases tumorigenicity, stemness, and chemoresistance in NSCLC cell lines through the antioxidant response system. Indeed, HNMT downregulation (obtained by overexpressing miR-223) sensitizes NSCLC to cisplatin chemotherapy [107].

In two other studies, Han and colleagues and Zhao and colleagues reported that miR-223 reverses resistance to erlotinib by directly targeting IGF-1R, and, thus, the PI3K/Akt signaling pathway, in the PC9-Erlotinib-resistant cell line they developed. Indeed, ectopic miR-223 expression in the PC9-Erlotinib resistant cell line enhances their sensitivity to erlotinib by inducing apoptosis, and inhibits tumor growth in nude mice [108,109].

#### 3.2.4. MiR-223 as Biomarker of NSCLC

Li and colleagues found that miR-223 levels in NSCLC tissue are inversely correlated with the patients’ overall survival [101].

MiR-223 is highly present in platelets and platelet-derived microvesicles under physiological conditions [110]. Liang and colleagues showed that platelet counts and the miR-223 levels in the platelets and platelet-derived MVs were increased in patients with NSCLC compared to healthy subjects [102].

The most abundant literature regarding miR-223 roles in NSCLC concerns its application as a minimally invasive diagnostic biomarker (Table 2), either alone or within a miRNA signature. Indeed, the literature on miR-223, by consistently evidencing its increased levels in biofluids and its ability to identify and discriminate between early- and late-stage cancers in patients, supports circulating miR-223 (either free in serum or plasma, or encapsulated in EVs, such as exosomes) as a good biomarker for NSCLC [111,112,113,114,115,116,117,118,119,120]. In some instances, miR-223 also distinguishes between cancer and other lung pathologies in patients [111,112,118]. Moreover, miR-223 is a good biomarker for both ADC and SCC NSCLC subtypes [116].

Wu and colleagues reported that either serum or serum exosomal miR-223 can distinguish early-stage NSCLC patients from healthy controls [118]. However, Chen and colleagues showed that, when whole plasma samples are used instead of EVs or EV-free plasma, miR-223 (together with other miRNAs) better distinguishes patients with lung ADC from those with granuloma [121].

MiR-223 levels are also higher in sputum from patients with NSCLC as compared to healthy controls. Among all tested sputum miRNAs, miR-223 performed the best as an NSCLC biomarker [122].

Circulating miR-223 decreases after surgery in patients with NSCLC, confirming its tumor-related or tumor-induced origin [114]. However, in some NSCLC cases diagnosed using miR-223 as a biomarker, the measured miR-223 levels did not change with surgery [123]; differences in post-surgery recovery time (and possible relapse) may explain these apparent discrepancies.

More recently, Monastirioti and colleagues tested the hypothesis that circulating miRNAs, which regulate immune checkpoints, regulatory T cells (Tregs), and Myeloid-derived suppressor cells (MDSCs), are prognostic biomarkers in patients with advanced NSCLC who are treated with anti-programmed cell death protein 1 (PD-1) (also called Nivolumab, a second-line anti-cancer drug). They found that low plasma miR-223 expression is negatively associated with prolonged disease control in patients with SCC, and such patients exhibit earlier disease progression than those having higher miR-223 expression [124].

Clearly, several aspects need to be addressed before miR-223 can be fully exploited as a lung cancer biomarker. Firstly, the source of circulating miR-223 in cancer patients must be thoroughly understood. Secondly, studies validating the ability of miR-223 to distinguish between lung cancer and benign lesions should involve larger cohorts of carefully selected subjects.

### 3.3. MiR-223 in Breast Cancer

Breast cancer is a complex and heterogeneous disease [125] characterized by signaling pathways involving multiple receptors that play a crucial role in its development and progression [126]. The implemented screening programs have resulted in a global increase in the reported invasive breast cancer cases [127]. Breast cancer is classified based on the combinational presence/absence of three receptors—estrogen receptor (ER), progesterone receptor (PR), and human epidermal growth factor receptor 2 (HER2). A tumor lacking all three receptors is defined as triple-negative breast cancer (TNBC). More recently, breast cancer has been classified as Luminal A, Luminal B, HER2-enriched, and basal-like based on the mRNA expression profile [128]. Despite several tests in use for clinical treatment and prognosis prediction [129], clinically relevant biomarkers or effective therapeutic targets of breast cancers are still lacking. Extensive research is ongoing to identify new biomarkers that can optimize personalized treatment, particularly for TNBC, which has a high risk of recurrence and lacks effective targeted therapy [130]. Surgical resection is the primary treatment for breast cancers; however, approximately 40% of patients experience recurrence or metastasis, leading to increased mortality rates. Diagnosis of breast ductal carcinoma in situ (DCIS) represents a risk factor for the development of invasive ductal carcinoma (IDC). This highlights the importance of identifying biomarkers that could classify DCIS patients as at “high risk” or “low risk” of developing IDC. Furthermore, despite advances, the underlying mechanisms leading to breast cancer development in most patients remain unclear. Hence, there is a critical need to identify new molecular markers and therapeutic targets to improve the survival rates of individuals affected by this disease.

#### 3.3.1. MiR-223 Role as an Onco-Suppressor in Breast Cancer

MiR-223 may suppress the development of breast cancer by targeting multiple oncogenic transcripts (Table 1), including epithelial cell transforming 2 (ECT2), Profilin 2 (PFN2) and NLRP3 [131,132,133]. Zhao and colleagues further demonstrated that miR-223 can be sponged by circABCB10, contributing to increased expression of PFN2 and to breast cancer progression, both in vitro and in vivo [133]. NLRP3 represents a potential novel target for breast cancer treatment. MiR-223 may influence tumor growth and immunosuppression in breast cancer by inhibiting the NLRP3 inflammasome [131]. Inflammasomes are multiprotein complexes that regulate various inflammatory factors, including IL-1β and IL-18. IL-18 induces immunosuppression in cancer through dependence on PD-1 [134], whereas IL-1β is a crucial inflammatory mediator with immunosuppressive properties [135]. In MCF-7 cells, NLRP3 expression increases following miR-223 inhibition. Knockdown of NLRP3 induces cell apoptosis in MCF-7 cells and reduces cell migration. These findings suggest that miR-223 may influence the progression of breast cancer cells by inactivating the NLRP3 inflammasome. Furthermore, in vivo studies demonstrate that miR-223 overexpression significantly decreases NLRP3 inflammasome expression, leading to reduced breast cancer growth [131]. Yang and colleagues suggested that the oncosuppressive role of miR-223 in breast cancer is also exerted through the targeting of stromal interaction molecule1 (STIM1), a known oncoprotein and calcium sensor protein in the endoplasmic reticulum which is involved in breast cancer progression and metastasization [136]. The authors found that breast cancer tissues overexpress STIM1 compared to the surrounding normal tissue and showed an inverse correlation between STIM1 and miR-223 expression in these tissues. They demonstrated that miR-223 inhibits the proliferative and migratory abilities of MDA-MB-231 and MCF7 cells by targeting STIM1 3′ UTR [136].

Intriguingly, TNBC stem cells exhibit low miR-223 expression compared to non-cancer stem cells (non-CSCs). Overexpression of miR-223 downregulates its direct target HCLS1-associated protein X-1 (HAX1) (an anti-apoptotic gene); sensitizes cancer stem cells (CSCs) to TRAIL-induced apoptosis via the mitochondria/ROS pathway; and subsequently activates the effectors caspase-9, caspase-7, and caspase-3 [137].

One notable circRNA called circZFR (circular RNA zinc finger RNA binding protein) is upregulated in breast cancer tissue and cell lines. CircZFR acts as an onco-circRNA in breast cancer by modulating the miR-223/fatty acid-binding protein 7 (FABP7) axis and reducing miR-223 activity. FABP7, a protein whose expression positively correlates with circZFR, crucially promotes EMT, migration, invasion, and proliferation of breast cancer cells [138]. FABP7 belongs to the intracellular lipid chaperone family and is typically expressed in the mammary glands. It is upregulated in TNBC and Her2-positive breast cancers compared to other breast cancer subtypes [139]. In patients with Her2-positive breast cancer, high FABP7 expression is associated with lower survival rates and a higher incidence of brain metastases [140].

Lipid metabolism is intricately linked to tumor progression, with several miRNAs, including miR-223, playing crucial roles in governing it. The cholesterol metabolite 27-hydroxycholesterol regulates breast cancer metastasis by remodeling the TME. MiR-223 expression inhibits lipogenesis, enhances cholesterol efflux by directly targeting and repressing scavenger receptor class B member 1 (SCARB1), and suppresses cholesterol uptake. Additionally, miR-223 inhibits cholesterol biosynthesis by directly repressing 3-hydroxy-3-methylglutaryl-CoA synthase 1 (HMGCS1) [141]. Moreover, miR-223 indirectly promotes ATP Binding Cassette Subfamily A Member 1 (*ABCA1*) expression, leading to enhanced cellular cholesterol efflux. Therefore, miR-223 holds potential as a biomarker for altered cholesterol homeostasis in breast cancer [142]. Cholesterol synthesis is also associated with an increase in cancer cell stemness, which is crucial for initiating metastasis [143].

#### 3.3.2. MiR-223 as an Onco-miRNA in Breast Cancer

In line with the dual role of miR-223, as described in other cancer types, miR-223 can also act as an oncomiR in the context of breast cancer (Table 1). MiR-223 is also associated with breast cancer progression and is found in exosomes shed from breast cancer cells [144], as well as in exosomes released from activated macrophages in breast cancer TME [52]. The latter exosomes contribute to miR-223 overexpression in breast cancer cells, consequently mediating the increase in their invasiveness through the negative regulation of the Mef2c-β-catenin pathway [52].

Du and colleagues showed that miR-223 may enhance the cell proliferation, migration, invasion, and EMT of breast cancer cells through the Hippo/Yes-associated protein 1 (Yap1) signaling pathway [145]. Inhibiting miR-223 expression increases Yap1 phosphorylation and large tumor suppressor kinase 1(LATS1) protein expression. LATS1 is a positive regulator of EMT, but no direct targeting has been described by the authors [145].

In addition, miR-223 controls Notch signaling [71,146], which also plays a significant role in human breast cancer development [147]. Under normal physiological conditions, Notch signal overactivation is prevented by a negative regulatory mechanism. Activated Notch is proteolyzed into the Notch intracellular domain (NICD) which translocates to the nucleus and interacts with transcription factor Recombination Signal Binding Protein for Immunoglobulin Kappa J Region (RBP-Jκ). NICD can be marked for ubiquitination via phosphorylation by CDK8 recruited to the NICD–RBP–Jκ complex. Subsequently, phosphorylated NICD is degraded by the E3 ligase FBXW7 [148]. Conversely, the loss of this negative regulation results in NICD stabilization and accumulation in the nucleus (a critical characteristic of overactive Notch signaling). This phenomenon has been widely implicated in tumor development and progression [149]. Generally, FBXW7 primarily exerts its tumor suppressor function by targeting a network of oncoproteins, such as mTOR, cyclin E, and Notch, for ubiquitination and proteasome degradation. FBXW7 deletion is frequently observed in human cancers, including breast cancer, and its loss contributes to tumorigenesis. Thus, miR-223 directly targets FBXW7, thereby exerting an oncogenic function in breast cancer, by increasing cell invasiveness and metastasization capability. Nude mice injected with “FBXW7 silenced MDA-MB-231-antimiR-223 cells” show a significantly higher number of lung and liver metastases than those injected with control cells [150].

#### 3.3.3. MiR-223 in Breast Cancer Drug and Radiotherapy Resistance

The role of miR-223 in breast cancer therapy resistance has been described, as reported in Table 1. In tamoxifen-resistant MCF7 breast cancer cells, genes associated with cholesterol metabolism are upregulated, and levels of cholesterol-rich lipid rafts containing various signaling proteins and receptors are elevated [151]. Palma and Kaur found that miR-223 overexpression and miR-128 inhibition, in combination with tamoxifen + acetyl plumbagin (the cholesterol-depleting agent) treatment, reduces the viability of long-term estrogen-deprived MCF-7 and MDA-MB-231 breast cancer cell lines and ameliorates their cholesterol-mediated drug resistance [142]. RT is the current conventional treatment modality for breast cancer [152]. Noteworthily, miRNAs have emerged as influential modulators of RT responses, with research indicating that ionizing radiations induce changes in miRNA expression profiles in both in vitro and in vivo models [153]. MiR-223 is induced by intraoperative RT in post-surgery human breast cancer tissue, in mouse mammary glands in vivo, and in breast epithelial (tumoral and non-tumoral) cells in vitro. The irradiation of breast cells specifically activates miR-223 promoter activity, culminating in the upregulation of miR-223 expression and the consequent downregulation of epidermal growth factor (EGF), one of its validated targets [154]. In mice subjected to a surgical procedure followed by mammary gland irradiation, miR-223 levels exhibited a dose-dependent increase after RT. Intriguingly, in patients with breast cancer, the surgical procedure alone did not impact miR-223 expression. Consistent results were observed in experiments with mammary epithelial cells in vitro, where the non-malignant MCF-10A cells expressed significantly higher levels of miR-223 compared to the basal-like breast cancer cell line MDAMB-231 [154]. The negative effect on cell viability after intraoperative RT was attributed to the targeting of EGF by miR-223. Even so, the authors suggest the possibility that breast cancer cells overexpressing miR-223 are sensitized to Lapatinib [154]. Conversely, Zhao and colleagues found that circABCB10 contributes to the radioresistance of breast cancer cells by sponging miR-223 [133]. These findings shed light on the potential regulatory role of miR-223 in the context of RT, providing valuable insights for future personalized approaches in breast cancer treatment.

#### 3.3.4. MiR-223 as Biomarker of Breast Cancer

Studies characterizing the exosomal miRNAome in patients with breast DCIS and IDC are limited. Notably, Yoshikawa and colleagues showed that, among exosomal miRNAs, miR-223 exhibits the highest increase in plasma-derived exosomes from IDC patients compared to those from healthy controls and DCIS patients [144]. Further, they suggest that miR-223 could serve as a promising biomarker (Table 2) for detecting IDC in patients initially diagnosed with DCIS [144].

### 3.4. MiR-223 in Ovarian Cancer

Ovarian cancer (OC) is one of the most aggressive types of neoplasm of the female reproductive system and ranks as the fifth leading cause of cancer-related deaths among women worldwide [54,155]. It can be classified into four subtypes: serous, endometrioid, clear cell, and mucinous ADCs [53,156].

Unfortunately, effective screening methods for early detection of OC are still lacking. Moreover, OC is often referred to as a silent cancer because its early stages typically do not exhibit symptoms, and the symptoms that appear in advanced stages are nonspecific. In fact, >75% of affected women receive their diagnosis when cancer is at advanced stages. Currently, diagnostic methods for detecting and monitoring epithelial OC (EOC) primarily involve pelvic examination, transvaginal ultrasound, and measurement of the serum biomarker carbohydrate antigen 125 (CA125). However, the detection of CA125 lacks sufficient specificity for early-stage diagnosis of OC, as elevated levels are observed in only 50% of stage I cases [157]. Moreover, the current treatment regimen of OC often leads to recurrence due to resistance to chemotherapy. To tackle these issues, the identification of new, early, and non-invasive diagnostic biomarkers, such as miRNAs, is necessary [158].

#### 3.4.1. MiR-223 as an Onco-microRNA in Ovarian Cancer

Fang and colleagues showed that miR-223 is upregulated in the OC cell lines SKOV3, OVCAR3, A2780, and ES2 compared to a non-neoplastic ovarian cell line. Moreover, ectopic miR-223 expression in OC cell lines increases the levels of Ki67 (marker of proliferation Kiel 67) protein and the proliferating cell nuclear antigen (PCNA). Conversely, knocking down miR-223 with a miR-223 inhibitor decreases Ki67 and PCNA levels. They also showed that specific inhibition of miR-223 in OC cells inhibits cell migration and invasion by relieving its direct targeting of SRY-box transcription factor 11 (SOX11). They further found that miR-223 inhibition increases SOX11 protein expression. Accordingly, SOX11 acts as a tumor suppressor in EOC [159]. Interestingly, high levels of SOX11 also promote epithelial–mesenchymal transition (EMT) in metastatic OC [160]. MiR-223 can be naturally sponged by circular RNA BNC2 (circBNC2/hsa_circ_0008732). In OC, circBNC2 is downregulated as compared to healthy tissue, and patients with lower circBNC2 expression exhibit a poorer prognosis. Ectopic expression of circBNC2 suppresses the viability, migration, invasion, and cell cycle progression of OC cells, while knockdown of circBNC2 has the opposite effect. Additionally, circBNC2 overexpression inhibits lung metastasis of OC cells in vivo, establishing circBNC2 as a novel onco-suppressor in OC [13]. More specifically, circBNC2 may act as an onco-suppressor in OC by sponging miR-223 and consequently regulating the miR-223/FBXW7 and miR-223/La-related protein 4 (LARP4) axes [13,161]. Downregulation of FBXW7, a gene involved in the regulation of the G1-S checkpoint of the cell cycle, favors the progression of EOC, particularly in stages III and IV, through the accumulation of oncoproteins such as Cyclin E and c-Myc [162]. In vivo studies have demonstrated how the downregulation of circBNC2 and LARP4, concomitant with the upregulation of miR-223, favors EOC growth [161]. Thus, the circBNC2/miR-223/LARP4 axis could potentially serve as a target regulatory mechanism for treating EOC. All the studies related to this section are shown in Table 1.

#### 3.4.2. MiR-223 in Ovarian Cancer Drug Resistance and Prognosis

Despite the currently available standard treatment options, the prognosis for patients with OC is still poor. The main reason for treatment failure is the development of chemoresistance, particularly against cisplatinum [51,163]. Tumor-associated macrophages (TAMs), which play a pivotal role in cancer progression by facilitating the transfer of exosomal miR-223, also regulate drug resistance. Zhu and colleagues, particularly, demonstrated that TAM-derived miR-223-enriched exosomes are internalized by EOC cells (especially under hypoxic conditions), which downregulates the miR-223 direct target *PTEN*, a known onco-suppressor gene and a key regulator of drug resistance. In EOC cells, decreased PTEN levels correlate with higher cisplatin-resistance, which is exhibited as decreased apoptosis levels and an increased cell viability rate brought about by the activation of PI3K/AKT pathway [51]. They also showed that increased miR-223 levels in TAM-derived exosomes correlate with shorter progression-free survival (PFS) of patients with OC [51]. Together, these studies, as reported in Table 1, support the role of miR-223 in ovarian cancer drug resistance and prognosis (Table 1).

#### 3.4.3. MiR-223 as a Biomarker of Ovarian Cancer

MiR-223 is upregulated in relapsed ovarian serous adenocarcinoma tissue as compared to the primary tumor, suggesting that it is a biomarker of recurrent OC [164]. Similar results were reported by Zhu and colleagues, who found a higher probability of recurrence (and consequent shorter PFS, see above) in patients with OC, showing increased miR-223 levels in serum-derived exosomes [51]. Fang and colleagues showed that miR-223 upregulation in OC tissue positively correlates with the presence of lymph node metastasis, histological tumor grade, and the International Federation of Gynecology Obstetrics (FIGO) stage [159]. The studies presented here and in Table 1 support the proposition of miR-223 as a biomarker of ovarian cancer.

### 3.5. MiR-223 in Prostate Cancer

Prostate cancer (PCa) is one of the most common cancers in men worldwide, and is considered one of the leading causes of cancer-related deaths [54]. PCa is a heterogeneous disease that can remain localized to the prostate or metastasize to distant organs. The incidence of PCa has increased over the last three decades, probably because of the improvement in diagnosis owing to the widespread use of PSA (prostate-specific antigen) screening and Gleason scoring systems. PSA screening, however, can lead to false positives, sometimes resulting in over-treatment without a real improvement in overall survival [165]. Since PSA is also produced in subjects with inflammatory lesions of the prostate (such as benign prostatic hyperplasia), its use as diagnostic biomarker alone is insufficient to identify patients with PCa.

MiRNAs have been investigated as additional alternative, minimally invasive diagnostic and prognostic biomarkers of PCa. Various miRNAs are relevant in PCa pathogenesis. In 2007, Porkka and colleagues performed the first extensive miRNA expression profiling of PCa cell lines, PCa murine samples, and clinical tumor samples. They identified 37 downregulated and 14 upregulated miRNAs that significantly supported PCa development [166]. Rana and colleagues, in their recent systematic review (covering 128 studies), identified miRNAs associated with tumor progression (let-7b-5p, miR-145-5p, miR-152-3p, miR-195-5p, miR-224-5p) and with recurrence (let-7a-5p, miR-148a-3p, miR-203a-3p, miR-26b-5p, miR30a-3p, miR-30c-5p, miR-30e-3p, miR-374a-5p, miR-425-3p, miR-582-5p) [167]. Here, we will discuss the role of miR-223 in PCa.

#### 3.5.1. MiR-223 as an Onco-Suppressor in Prostate Cancer

MiR-223 is sponged by circRNA G protein subunit γ 4 (circGNG4), resulting in increases in tumor-promoting factor EYA3 (a direct target of miR-223) and, in turn, c-Myc levels. This enhances the malignant behavior of the PCa cell line PC-3, both in vitro and in vivo (in tumor-bearing mice) [168]. Thus, it is suggested that miR-223 is an onco-suppressor in PCa. The studies discussed here are also summarized in Table 1 and Figure 1.

#### 3.5.2. MiR-223 as an Onco-miRNA in Prostate Cancer

Wei and colleagues showed that miR-223 is upregulated in PCa tissue compared to healthy adjacent tissue, and is also increased in poorly differentiated tumors as compared to their well-differentiated counterparts [169]. In the PCa cell lines DU145, PC3, and LNCaP, miR-223 inhibition impairs cell proliferation by inducing apoptosis, inducing cell cycle arrest in G0/G1, and also limiting cell migration [169]. Mechanistically, the authors identified septin 6 (*SEPTIN6*) as a direct target of miR-223 by demonstrating that septin 6 overexpression rescues the phenotype induced by miR-223 in PCa cells [169]. This study (Table 1 and Figure 2) supports the role of miR-223 as an onco-microRNA in prostate cancer.

#### 3.5.3. Role of miR-223 in Drug Resistance and Prognosis of Prostate Cancer

Feng and colleagues showed that miR-223 inhibition induces apoptosis and suppresses cell survival in both the hormone-resistant DU145 and the hormone-sensitive LNCaP PCa cell lines [170], as mechanisms of drug resistance (Table 1). They studied the impact of miR-223 on the in vitro and in vivo growth of PCa cells in response to chemotherapeutic docetaxel treatment. They found that high miR-223 levels are associated with chemoresistance to docetaxel. Indeed, inhibiting miR-223 sensitizes PCa cells to docetaxel treatment via upregulating its direct target forkhead box 3 (FOXO3) transcription factor [170]. In agreement with Feng and colleagues’ findings, Zhou and colleagues showed that high miR-223 levels reduce the radio-sensitivity of PC3 and LNCaP cells by targeting FOXO3, leading to the activation of glycolysis [171]. In PCa, miR-223-mediated modulation of the Warburg effect (that is the production of energy through the anaerobic glycolysis also in aerobic conditions) via FOXO3a inhibition reduces radio-sensitivity [172].

#### 3.5.4. MiR-223 as Biomarker of Prostate Cancer

Some studies support the feasibility of using miR-223 as a biomarker of PCa (Table 2). Dülgeroğlu and colleagues found that serum miR-223 and miR-223-5p levels decrease in patients with PCa or chronic prostatitis as compared to those affected by benign prostate hyperplasia (BPH) [173]. Considering the roles of miR-223 in the closely linked, complementary inflammation and carcinogenic processes, further studies are required in order to better identify whether miR-223’s role is restricted to cancer-promoting inflammation (CPI) or classical chronic inflammation. Zhou and colleagues found increased miR-223 levels in urine samples from patients with PCa who were radio-resistant as compared to those who were radio-sensitive, suggesting the utility of this miRNA as a biomarker of the response to RT [171].

### 3.6. MiR-223 in Glioma and Glioblastoma

Malignant gliomas and glioblastoma (GBM) are the most common cancers of the central nervous system (CNS), constituting > 80% of all CNS malignancies. Among CNS malignancies, GBM is the most common and most lethal, with a median survival of 15 months from diagnosis, despite the Stupp protocol-based radical treatment (used since 2005) [174,175]. The recent WHO 2021 classification of CNS tumors separated GBM from gliomas; therefore, from a histological and molecular point of view, GBM represents its own rightful entity. In turn, gliomas can be classified into low-grade (grades I and II; slow-growing tumors with a generally favorable prognosis) and high-grade (grade III; fast-growing, infiltrating tumors with a generally poor prognosis) [176]. Although, recently, the molecular basis of gliomas and GBM has been increasingly clarified [177,178,179,180,181,182,183], biomarkers representing a valid diagnostic, prognostic, or therapeutic signature are still lacking. As in other tumors, the expression of several miRNAs is altered in glioma and GBM cells compared to unaffected brain parenchyma. In several cases, the dysregulated expression of these miRNAs matches with their functional involvement, both in terms of variation in the expression of their targets as well as their effects on the neoplastic phenotype [17,184,185]. More recently, the contribution of other non-coding RNAs, such as lncRNAs and circRNAs, to regulating miRNA-mediated gene expression modulation has been recognized [12].

In this context, miR-223 is differentially expressed in gliomas and GBM compared to normal tissue, and it regulates cell growth, proliferation, inflammation, and response to chemotherapy through several molecular mechanisms.

#### 3.6.1. MiR-223 as an Onco-Suppressor in Glioma and GBM

Glasgow and colleagues were the first to suggest that miR-223 is an onco-suppressor in CNS tumors, particularly in GBM, by performing studies on chicken embryogenesis [186]. This was followed by other studies, as detailed in Table 1. They showed that miR-223 negatively regulates nuclear factor 1A (NFIA) expression, contributing to the differentiation of glia and neuron precursors. Furthermore, miR-223 and NFIA expression levels are negatively correlated in human GBM biopsies. In addition, the GBM cell line U87, which ectopically expresses miR-223, proliferates significantly less than the control cell line because of the G1 phase cell cycle arrest caused by p21 onco-suppressor activation. The onco-suppressive activity of miR-223 was also highlighted by Evers and colleagues, who showed that it is overexpressed in glioma stem cells (GSCs) isolated from GBM patients when they are induced to differentiate. In particular, the authors focused on a subpopulation of chemoresistant GSCs which efficiently secreted solutes due to the aberrant activity of ABC transporters. The authors accurately predicted that miR-223 would directly target the stemness marker Musashi RNA-binding protein 2 (MSI2) within this subpopulation; thus, miR-223 upregulation can contribute to GSC differentiation [187].

In turn, miR-223’s function is regulated upstream by at least two lncRNAs, SNHG29 and PITPNA-AS-1, which are both upregulated in GBM. Han and colleagues demonstrated how SNHG29, through a ceRNA mechanism, can act as a molecular sponge for miR-223. Thus, SNHG29 can activate delta 1 catenin (CTNND1), a direct target of miR-223, resulting in increased cell proliferation, migration, and EMT via the Wnt/β-catenin pathway [188]. Geng and colleagues demonstrated that the onco-suppressive activity of miR-223 is blocked by PITPNA-AS-1 in GBM cells. They proposed that PITPNA-AS-1 acts as a molecular sponge for miR-223, enabling cell proliferation, and significant decrease in apoptosis through upregulation of EGFR (a direct target of miR-223) and activation of the PI3K/AKT pathway [189].

Through a study conducted on GBM biopsies and cell lines, Ding and colleagues showed that miR-223 acts as an onco-suppressor by inhibiting cell proliferation and migration by downregulating its direct target, NLRP3 (involved in the inflammasome formation), and, consequently, the NLRP3-associated pro-inflammatory cytokines (IL-1β, MCP-1, IL-18 and IL-8) and the pro-inflammatory mediator caspase 1 [190].

#### 3.6.2. MiR-223 as an Onco-miRNA in GBM

Some studies have reported an oncogenic role of miR-223 in GBM (Table 1). Zhang and colleagues showed how miR-223 expression positively correlates with the expression of genes involved in the EMT [191]. Moreover, ectopic expression of miR-223 in GBM cell lines increases their invasiveness. Similarly, Huang and colleagues observed that several GBM cell lines overexpress miR-223 as compared to the human fetal glial cell line HFGC. Further, they also showed reduced expression of its onco-suppressor target paired box 6 (PAX6); GBM tissues also showed increased miR-223 and reduced PAX6 expression compared to control tissues during an immunohistochemical characterization on a tissue microarray [192]. Furthermore, they found that miR-223 inhibition downregulated the GBM progression genes which were repressed by PAX6 (such as the metalloproteinases MMP2 and MMP9 and vascular endothelial growth factor VEGFA). In parallel, they also demonstrated that anti-miR-223 decreases cell viability and invasiveness. In a subsequent study, Huang and colleagues evaluated the expression of miR-223 and PAX6 in the stem cell component (CD133^+^) and the non-stem cell component (CD133^−^) of the GBM cell line U251. They found that miR-223 was upregulated and PAX6 was downregulated in the CD133^+^ glioma stem cells (GSCs) compared to the CD133^−^ cells [193].

#### 3.6.3. MiR-223 in Drug Resistance and Prognosis of Gliomas and GBM

Huang and colleagues also demonstrated that increased miR-223 expression correlates with GSC chemoresistance to temozolomide (TMZ), caused by decreased PAX6 expression and consequent PI3K/AKT pathway activation [193]. Similarly, in a parallel study, Cheng and colleagues demonstrated that inhibiting miR-223 reduces the chemoresistance of glioblastoma (GBM) cells to temozolomide (TMZ) treatment [194]. Conversely, but in accordance with the onco-suppressive function of miR-223 in GBM, Liang and colleagues demonstrated that miR-223 overexpression sensitizes the U87 GBM cell line to radiation-induced cytotoxicity by targeting the *Ataxia telangiectasia mutated* (*ATM*) gene transcript [195]. They further showed that the post-RT tumor volume reduction seen in xenograft mice implanted with U87 cells overexpressing miR-223 was significantly higher as compared to that seen in xenograft mice implanted with U87 control cells [195]. Mekala and colleagues showed that N-acetyl L-aspartate and Triacetin (two oncosuppressive acetates) induced the expression of a battery of oncosuppressive miRNAs, including miR-223, thus increasing the chemosensitivity of GBM cells and inducing their apoptosis [196]. The studies discussed here are also summarized in Table 1.

#### 3.6.4. MiR-223 as Biomarker of Gliomas and GBM

In agreement with the onco-suppressive function of miR-223, several authors have suggested that its expression positively correlates with longer survival of patients with GBM [197,198,199,200,201,202]. For instance, Wang and colleagues showed that miR-223 is part of a group of five miRNAs whose tissue expression positively correlates with overall survival of patients with glioma and GBM. These miRNAs are united by having targets that positively regulate processes such as neuron migration, response to transforming growth factor, and pro-survival pathways such as MAPK signaling [197]. Similarly, Li and colleagues described miR-223 as a protective miRNA in the GBM mesenchymal subtype [198]. Conversely, Huang and colleagues described higher expression of miR-223 (together with a cancer tissue-derived gene expression signature of 16 mediator protein coding-genes) as negative prognostic factors in GBM [199]. Interestingly, Lu and colleagues identified miR-223 (together with 5 other miRNAs) as a prognostic factor in low-grade gliomas, but found that its higher expression in GBM tissue represented a risk factor [200]. In agreement with Lu and colleagues’ findings, miR-223 levels in whole pre-operative blood samples from patients diagnosed with diffuse low-grade glioma were higher than those in healthy controls [201]. In a previous study, Roth and colleagues also found a higher miR-223 level in whole blood samples from patients with GBM compared to those from healthy controls [203]. Serum-derived miR-223 levels also correlated with the actual progression of low-grade gliomas during post-surgery follow-up, as demonstrated by magnetic resonance imaging results [202].

In conclusion, the biological role of miR-223 in GBM is debated, with most authors agreeing on its onco-suppressive and protective role (in terms of patient survival) and others observing an oncogenic and high-risk group-related role in GBM patients. These discrepancies may arise from various causes, such as: (i) the subtype of tumor analyzed (i.e., proneural, neural, mesenchymal, or classical phenotype), as evidenced by Glasgow and colleagues and Genovese and colleagues [186,204], or (ii) the percentage of cancer stem cells in the samples analyzed, as observed by Huang and colleagues, who showed differences in miR-223 expression in GSCs compared to the non-stem component [194]. In general, however, the bibliographic sources analyzed herein indicate a growing interest in this miRNA; the pathways it regulates in brain tumors; and its use as a non-invasive prognostic, diagnostic, and therapeutic biomarker, as well as a therapeutic target (Table 2).

### 3.7. MiR-223 in Pancreatic Cancer

Pancreatic cancer (PC) is very invasive, has a poor prognosis, and exhibits high drug resistance [205,206,207]. In addition, PC is characterized by extensive fibrosis, which limits proper drug delivery to the tumor site and reduces the infiltration of effector immune cells (i.e., NK cells, CD8^+^ T cells) [208,209].

PC comprises two major subtypes: (1) exocrine and (2) neuroendocrine. Exocrine PC is more frequent than neuroendocrine PC [210]. Pancreatic ductal adenocarcinoma (PDAC) is the most prevalent exocrine PC, and is therefore highly studied [211]. Moreover, most PC cell lines available for research are derived from human PDAC [212]. Thus, the molecular mechanisms leading to PDAC are fairly well understood. The genetic signature of the initial stages of PDAC is known. A combination of mutations in the genes—*KRAS proto-oncogene*, *GTPase* (*KRAS*), *Cyclin-Dependent Kinase Inhibitor 2A* (*CDKN2A*), *Tumor Protein p53* (*TP53*), *Lysine Demethylase 6A* (*KDM6A*), *BRCA1* and *BRCA2 DNA repair-associated* (*BRCA1 and 2*), and *SMAD Family Member 4* (*SMAD4*)—are involved in PDAC genesis [212,213].

Many miRNAs are perturbed in PC cells, early PC lesions, or patient plasma and urine samples [214,215]. Therefore, they could serve as diagnostic or prognostic biomarkers of PC. MiRNAs modulated in PC have been extensively reviewed elsewhere [216,217,218]. Here, we focus on miR-223 modulation and its roles in PC (Table 1). Evidence for miR-223 involvement in PC is still emerging.

#### 3.7.1. MiR-223 as an Onco-miRNA in PC

Studies on miR-223 as an Onco-microRNA in PC are described here and in Table 1. Rachagani and colleagues showed that, in an in vivo PC progression model (*Kras*^G12D^; Pdx1-Cre (KC/floxed Kras^G12D^ mice)), miR-223 expression progressively increased over time, reaching its maximum level at the 30th week of development, immediately before the complete transformation of pancreatic intraepithelial neoplasia lesions into PDAC [219]. Further, they confirmed the upregulation of miR-223 in human PC tissues as compared to the cancer-adjacent normal tissues, suggesting an oncogenic role of this miRNA in PC. Other studies that point to an oncogenic role of miR-223 in PC show that miR-223 mainly targets two known onco-suppressors, PDS5 cohesin-associated factor B (PDS5B/APRIN) and FBXW7 [220,221,222]. Below, we will discuss how inhibition of these targets contributes to PC pathogenesis and progression.

MiR-223 directly targets PDS5B mRNA, causing an increase in cell growth, migration, and invasion and a decrease in apoptosis in vitro in the PC cell lines AsPC-1, PANC-1, and PaTu-8988 [220]. MiR-223 also favors in vivo tumor growth, as seen in the tumors formed in nude mice by the injected PaTu-8988 cells, which ectopically expressed miR-223 [220]. PDS5B is an established onco-suppressor whose downregulation inhibits cell differentiation by disrupting sister chromatid cohesion, causing “cancer-initiating cells (CICs)” to accumulate [223,224]. Further, PDS5B depletion enhances the proliferation of CICs by activating the IL6/STAT3/CCND1 axis [223]. PSD5B depletion also downregulates patched 2 (PTCH2) expression, thereby activating the Sonic hedgehog pathway, which in turn enhances PC cell growth, migration, and invasion [225]. Interestingly, PDS5A and PDS5B are paralogs of PDS5; they have redundant and specific functions [223]. Thus, it would be interesting to see if miR-223 also affects PDS5A expression and if PDS5A has roles in PC.

He and colleagues confirmed the oncogenic role of miR-223 in PDAC; they found that miR-223 is upregulated in PDAC tissue as compared to adjacent healthy tissue, and that its overexpression increases cell proliferation and migration, decreases in vitro apoptosis, and enhances in vivo tumor growth [221]. Mechanistically, miR-223 exerts its oncogenic function by directly targeting the onco-suppressor FBXW7, which in turn negatively controls the expression of the heterogeneous nuclear ribonucleoprotein K (hnRNPK), an activator of pri-miR-223 transcription. This creates a positive feedback loop that leads to further enhancement of the hnRNPK/miR-223/FBXW7 pathway [221].

Zhang and colleagues showed that miR-223 negatively regulates the expression of SLC4A4 via direct targeting [226]. Further, they showed that human circular RNA 001587 (hsa_circ_0000979 or hsa_circ_001587) sponges miR-223, thus relieving its negative regulation of SLC4A4. Indeed, physiological or pharmacological inhibition of miR-223 reduces PC cell migration, invasion, angiogenesis, and tumorigenesis in vitro, as well as tumor growth in vivo [226]. Although, in prostate cancer, SLC4A4 is oncogenic [227], in renal cancer and PC, it behaves as a tumor suppressor [226,228].

#### 3.7.2. Role of miR-223 in Drug Resistance and Prognosis of PC

Different studies discussed here and shown in Table 1 support the role of miR-223 in drug resistance and prognosis of PC (Table 1). Huang and colleagues found that miR-223 is upregulated in the cisplatin-resistant BXPC3/CDDP PC cell line compared to the cisplatin-sensitive BXPC3 counterpart [229]. They also found that miR-223 is functionally involved in cisplatin resistance through direct targeting of FOXO3 and the consequent increase in cell proliferation and decrease in apoptosis [229]. MiR-223 inhibition increases FOXO3 levels and restores cisplatin sensitivity [229]. Active FOXO3 is proapoptotic and antiproliferative [230]. In early PC tumors, FOXO3 is upregulated, but is progressively downregulated in advanced stages [231]. Moreover, low expression of FOXO3 correlates with decreased survival of patients [231]. FOXO3 further promotes the Warburg effect in PC cells; as reported in the pertinent section, in PCa, miR-223 modulation of the Warburg effect via FOXO3a inhibition reduces radio-sensitivity [172]. The effects of the miR-223/FOXO3 axis in modulating glycolysis and radiosensitivity in PC remain to be studied and may represent an interesting field of study. Also, gemcitabine-resistant PDAC cell lines have increased levels of miR-223 [232]. Reversal of miR-223 overexpression in gemcitabine-resistant PDAC cell lines through treatment with genistein (an anticancer soy isoflavonoid) and miR-223 inhibitors upregulates FBXW7 (the direct target of miR-223) in vitro and in vivo [222,233]. FBXW7 allows the turning off of Notch-1 signaling pathway, consequently leading to EMT suppression [222,232].

#### 3.7.3. Role of miR-223 as a Biomarker in PC

MiR-223 levels are increased in the whole blood, plasma, and serum of patients with PC as compared to healthy controls. Moreover, plasma miR-223 levels decrease after surgery in patients with PC, suggesting a direct link between its expression and PC tumor presence [234,235] and supporting its utility as a biomarker in PC (Table 2). Wang and colleagues also suggested plasma-derived miR-223 as a prognostic biomarker of the malignant behavior of intraductal papillary mucinous neoplasm (IPMN), independent of tumor size [236]. They also reported a significant upregulation of miR-223 in IPMN tissue as compared to normal pancreatic tissue [236]. MiR-223 levels are elevated in urine samples from patients with stage I PDAC compared to those from patients with Stage II-IV PDAC and healthy subjects [237]. Combined with other miRNAs, miR-223 could be used to identify patients with early-stage PDAC [235,237]. MiR-223, in combination with miR-204, can distinguish between early PDAC and chronic pancreatitis [237].

Conversely to what has been reported in PDAC, miR-223 plasma levels are lower in patients with neuroendocrine tumors (NET) as compared to healthy controls [238]. Further, miR-223 does not have prognostic value or reveal any association with histopathological characteristics in patients with NET [238,239]. Notably, miR-223 promotes the dedifferentiation of pancreatic islet β cells [240]. Islet β cells express miR-223. Dedifferentiation followed by redifferentiation is required for islet β cell proliferation in response to high glucose and TNFα exposure [240]. High miR-223 expression during type-2 diabetes is therefore considered a compensatory mechanism. FOXO1, Sox6, Pdx-1, P27, and Glut2 are involved in this process [240]. It would be interesting to see whether these molecules are involved in the genesis of neuroendocrine PC.

### 3.8. MiR-223 in Hematological Malignancies

Hematological malignancies are a very heterogeneous group of diseases arising from disrupted hematopoiesis [241]. They are mainly classified as myeloid and lymphoid tumors based on their cells of origin, and as acute or chronic depending on their aggressiveness. For recent classification of hematological malignancies, refer to the International Consensus Classification [242,243] and the World Health Organization Classification, 5th edition (WHO5) [244,245]. As mentioned in Section 2, miR-223 is mainly involved in hematopoiesis. Therefore, here, we will discuss its roles in hematopoiesis before discussing its roles in hematological malignancies (Table 1).

#### 3.8.1. MiR-223 in Hematopoiesis

In 2004, Chen and colleagues first reported that miR-223 is nearly exclusively expressed in bone marrow, where it contributes to the generation of several hematopoietic cells [246]. Fazi and colleagues reported that overexpressing miR-223 promotes myeloid differentiation [39]. Additionally, knocking down miR-223 in cord-blood CD34⁺ cells delays the differentiation of myeloerythroid precursors in vitro, but leads to an increase in myeloid progenitors after serial xenotransplantation in vivo. Furthermore, overexpressing miR-223 promotes erythropoiesis, early B lymphopoiesis, and T lymphopoiesis in vivo. Therefore, miR-223 plays a role in balancing the expansion and differentiation of human hematopoietic progenitors [247].

MiR-223 is transcriptionally regulated by NFIA, C/EBPα, PU.1, E2F1, and AML1-ETO fusion protein [248,249]. Functionally, it participates in a regulatory circuitry alongside C/EBPα and NFIA, which govern human granulopoiesis. Significantly, the induction of NFIA enhances erythropoiesis, while its repression leads to granulopoiesis. Its expression is inversely correlated with that of miR-223. The latter negatively controls *NFIA* expression at both the transcriptional and post-transcriptional levels [39,250]. MiR-223 expression is also modulated in erythroid cells. A study found that hemin-induced erythroid differentiation of K562 cells results in the downregulation of miR-223, whereas phorbol myristate acetate-induced megakaryocytic differentiation leads to its upregulation [251]. This study also confirmed that miR-223 regulates erythroid and megakaryocytic differentiation. Indeed, ectopic expression of miR-223 negatively regulates LIM domain-only 2 (LMO2) and reversibly regulates erythroid and megakaryocytic differentiation in K562 cells in a feedback loop [39,252].

Taken together, these data indicate that miR-223 expression is tightly modulated in hemopoietic cells. Its levels determine the lineage progression and the ratio among sub-cell populations. Hence, it is natural to expect that its dysregulation may lead to disrupted hematopoiesis and, thus, carcinogenesis.

MiR-223 acts mainly as an onco-suppressor in lymphoid and myeloid cancers. However, the pathways involved vary between acute (acute lymphocytic leukemia, ALL, and acute myeloid leukemia, AML) and chronic (chronic lymphocytic leukemia, CLL, and chronic myeloid leukemia, CML) types. Further, in some instances of AML and ALL, it acts as an onco-miRNA. In the next few sections, we will report the roles of miR-223 as an onco-suppressor and onco-miRNA in hematological malignancies in more detail. We will also discuss the pathways that affect miR-223 levels in these malignancies.

#### 3.8.2. MiR-223 as an Onco-Suppressor in Hematological Malignancies

##### MiR-223 as an Onco-Suppressor in ALL and AML (Table 1)

Zhang and colleagues found that, compared to healthy patients, miR-223 expression was downregulated in ALL patients [253]. Although it is downregulated in both ALL and AML, miR-223 expression is lower in ALL than in AML [254,255]. However, a subset of patients with T-cell ALL (exhibiting higher myeloid-like features) had miR-223 expression comparable to that in AML patients, higher than that in T-cell ALL patients and CD2^+^ cells from healthy donors [256].

A 3′UTR polymorphism, rs4946936 in FOXO3 (TT/CT vs CC allele), is associated with an increased risk of incurring childhood ALL in the Han Chinese population [257]. The T-alleles affect the binding affinity of miR-223 to FOXO3 3′UTR. Thus, FOXO3 mRNA expression is relatively higher in patients carrying TT/CT alleles as compared to those carrying the CC allele [257]. This confirms that miR-223 regulates the FOXO family of transcription factors.

In AML, miR-223 acts as an onco-suppressor by negatively regulating FBXW7. Reduced miR-223 expression increases the levels of the FBXW7 target, thus decreasing apoptosis and increasing cell proliferation [258]. Pulikkan and colleagues also reported the onco-suppressive function of miR-223 in AML, and they showed that its down-regulation contributes to the overexpression of the positive regulator of cell cycle E2F1. The latter, in turn, acts as a repressor of the transcription of the *MIR223* gene, thus creating a feedback loop that leads to decreased granulocytic differentiation and increased myeloid cell proliferation [259].

##### Oncogenic Pathways Which Act by Lowering miR-223 Levels in ALL and AML

Diverse mechanisms downregulate miR-223, causing predisposition to cancer development. MiR-223 downregulation can be associated with copy number alterations (CNA) and copy number neutral loss of heterozygosity (CNN LOH) in chromosome Xq12, as seen in ALL patients [260]. Other mechanisms involve upstream regulators of miR-223 expression. In various human leukemias (such as AML and infant/childhood ALL), expression of mutated FLT3, harboring internal tandem duplications of the juxtamembrane regions (FLT3/ITD), constitutively activates cell proliferation and survival pathways and is associated with poor prognosis. Ectopic expression of FLT3/ITD in normal bone marrow murine FDC-P1 cells downregulates miR-223 expression [261]. Eyholzer and colleagues showed that miR-223 expression is mainly regulated at the transcriptional level by CEBP transcription factors (CEBPα and β), which recognize a binding site upstream of the pre-miRNA sequence. Loss of CEBP function in AML patients may, thus, be responsible for miR-223 downregulation [262]. In line with these findings, Garzon and colleagues found that miR-223 is upregulated upon all-trans-retinoic acid (ATRA) treatment, in both NB4 and HL60 acute promyelocytic leukemia cell lines as well as in primary blast cells from acute promyelocytic leukemia patients. The authors hypothesize that miR-223 is upregulated due to concurrent downregulation of NFIA, the main competitor of CEBPα which, in turn, is targeted by the ATRA-induced miR-107 [263]. The *RUNX1* (*AML1*) gene encodes for one of the two subunits of the heterodimeric core binding factor (CBF), a hematopoietic-specific transcription factor. In AML cells, RUNX1 is often rearranged to form fusion proteins that negatively regulate the expression of miR-223, contributing to the proliferation of the undifferentiated myeloid cells [264].

Other factors expressed in cancer cells also regulate miR-223 levels in ALL and AML. For instance, the coactivator-associated arginine methyltransferase 1 (CARM1/PRMT4) is a type I arginine methyltransferase that blocks myeloid differentiation of human stem/progenitor cells (HSPCs) [265]. This protein is highly expressed in AML patients, where it suppresses miR-223 expression by methylating RUNX1 at arginine 223 residue, and thus determines the assembly of the double PHD fingers 2 (DPF2)-containing repressor complex on the *MIR223* gene promoter. The same authors demonstrated that PRMT4 is a validated target of miR-223, and that its depletion in vitro stimulates myeloid leukemia cell differentiation while decreasing their proliferation [265].

The environmental carcinogen hydroquinone (HQ) is the main metabolite of benzene which causes leukemia and lymphoma. During the later stages of HQ-induced transformation of the TK6 lymphoblastoid cell line, PARP-1 and miR-223 levels are up- and down-regulated, correspondingly, while apoptosis is decreased [266]. In this model, *MIR223* transcription is regulated by the PARP-1 and HDAC2 complex via histone acetylation [266].

Human T-cell lymphotropic virus type 1 (HTLV-1) causes adult T-cell leukemia (ATL). HTLV-1 induces the aberrant expression of several miRNAs, including miR-223. MiR-223 is downregulated in HTLV-1–infected cells in vitro, while it is upregulated in primary ex vivo ATL cells [267]. In human T-cells, miR-223 targets STAT1, thus dampening the STAT1-dependent signaling required for T cell proliferation [268]. Thus, HTLV-1 may promote ATL development by inactivating miR-223 function.

##### MiR-223 as an Onco-Suppressor in CLL and CML

Davari and colleagues showed that miR-223 is downregulated in the peripheral blood mononuclear cells of patients with CLL compared to healthy controls. Moreover, in the same cohort, miR-223 expression was inversely correlated with the age of the patient and the count of white blood cells; it decreased in smokers compared to non-smokers and was inversely correlated to STAT3 expression, its direct target [269]. Agatheeswaran and colleagues reported that MEF2C and polypyrimidine tract-binding protein 2 (PTBP2) are direct targets of miR-223, and are upregulated in CML. More specifically, they showed that the fusion protein BCL-ABL contributes to the downregulation of miR-223 by inhibiting C/EBPα translation, leading to increased myeloblast proliferation and to the aberrant splicing of several transcripts observed during CML progression [270]. Low expression of miR-223 also leads to high FLT3 expression in CML. CML cell lines co-treated with 2-methoxyestradiol (an endogenous estrogen metabolite) and ascorbic acid (an antioxidant) showed increased miR-223 and decreased FLT3 expressions, which consequently induced apoptosis by blocking the PI3K/AKT pathway; the co-treatment also inhibited tumor growth in a mouse model [271]. Together, the studies discussed here and shown in Table 1 outline the oncosuppressor properties of miR-223 in CLL and CML.

#### 3.8.3. Role of miR-223 as an Onco-miRNA in Hematological Malignancies

Paralleling what is observed in solid cancers, miR-223 can have divergent functions in hematological malignancies, acquiring oncogenic properties (Table 1). In normal monocytes (and during M1 polarization), high levels of monocytic leukemia zinc-finger protein (MOZ) repress miR-223 expression through the binding of PU.1 elements to the *MIR223* promoter. In acute monocytic leukemia (AMoL), the miR-223 increase negatively regulates its direct target MOZ through an autoregulatory loop mechanism, thus blocking monocyte differentiation and promoting disease progression. Low MOZ expression in the AML cell lines THP-1 or U937, obtained by MOZ knockdown or miR-223 ectopic expression, inhibits monocyte differentiation, maintains stemness, and induces resistance to cisplatin-induced apoptosis [272]. Thus, miR-223 acts as an onco-microRNA in AML and AMoL.

Multifunctional β-arrestin (ARRBs) adapter proteins are tumor suppressors in T-cell acute lymphoblastic leukemia (T-ALL). In T-ALL human samples, arrestin β 1 (ARRB1) is downregulated, and its ectopic expression in Jurkat T-ALL cells inhibits proliferation in vitro compared to control cells; further, the survival of nude mice injected with ARRB1 expressing cells is increased [273]. MiR-223 acts as onco-miRNA by suppressing ARRB1 translation, thereby enhancing NOTCH1 activity and T-ALL cell proliferation while reducing apoptosis [273]. Interestingly, ARRB1 acts as oncogene in B-cell acute lymphoblastic leukemia (B-ALL) [274,275]. It would be interesting to investigate the correlation between miR-223 and ARRB1 in this tumor to define similarities and differences with respect to T-ALL. Mir-223 also acts as a onco-miRNA by downregulating the aforementioned FBXW7 in T-ALL cells. MiR-223 may increase in T-ALL cells because of low expression of its natural sponge, circ_0000094. Circ_0000094 ectopic expression inhibits proliferation, migration, and invasion and induces apoptosis, possibly by increasing FBXW7 [276]. Kumar and colleagues showed that Notch3 and NF-κB co-regulate miR-223 expression by directly binding to its promoter in T-ALL cells; upregulation of miR-223 by Notch3 and NF-κB signals downregulates FBXW7 [277]. Braun and colleagues found that T-cell prolymphocytic leukemia (T-PLL) cells could be clustered based on miRNA expression profiles. The clusters expressing high miR-223 were associated with a more aggressive phenotype. Through gene set enrichment analysis (GSEA), they found that miR-223 was associated with DNA damage response (p53 pathway) and deregulated cell-cycle mediator signatures [278].

The oncogenic transcription factor TAL1/SCL is aberrantly expressed in human T-cell ALL (T-ALL). TAL1, along with its regulatory factors HEB, E2A, LMO1/2, GATA3, and RUNX1, directly controls miRNA genes, including *MIR223*, via direct promoter binding [279]. MiR-223 and TAL1 levels are correlated in human T-ALL during normal thymic development. TAL1 negatively regulates FBXW7 tumor suppressor through miR-223 upregulation. This has been further confirmed by TAL1 knockdown experiments, where increased FBXW7 and reduced c-Myc, Myb, Notch1, and Cyclin E (FBXW7-oncoprotein substrates) levels are observed [279].

#### 3.8.4. Role of miR-223 in Prognosis and Drug Resistance of Hematological Malignancies

In 2011, Han and colleagues carried out a genome-wide miRNA microarray analysis of paired diagnosis–relapse or diagnosis–complete remission (CR) bone marrow samples of childhood ALL, revealing miR-223 dysregulation [280]. Data analysis demonstrated that miR-223 is significantly downregulated in patients with relapsed ALL compared to those with CR. Moreover, miR-223 is upregulated in patients with CR, as compared to the levels at first diagnosis. Lower expression of miR-223 at first diagnosis is predictive of lower relapse-free survival for patients with ALL [280,281].

Kumar and colleagues showed that inhibiting miR-223 prevents T-ALL cells from developing resistance to γ-secretase inhibitor (GSI) treatment, indicating that miR-223 could be a target to induce GSI sensitivity [277]. Conversely, Gusscott and colleagues showed that miR-223 is upregulated in T-ALL after treatment with GSI. MiR-223 upregulation, in turn, causes downregulation of its direct target IGF1R in both sensitive and resistant cell lines, without any effect on cell survival [282].

Glucocorticoids (GC) induce apoptosis in cells of the lymphoid lineage and are used to treat ALL and related malignancies. MiR-223 expression is induced in a subset of pediatric ALL and some ALL cell lines sensitive to GC [283]. The role of mir-223 in prognosis and drug resistance in hematological malignancies is summarized in Table 1.

#### 3.8.5. Role of miR-223 as a Biomarker in Hematological Malignancies

Yu and colleagues observed that reduced serum miR-223 levels in patients with AML correlated with its aggressiveness and a poorer prognosis. Further, miR-223 levels increased post-treatment in patients with AML. Thus, miR-223 has been proposed as a diagnostic and prognostic AML biomarker [284]. Given that the miR-223 level is lower in the bone marrow from patients with ALL compared to that from patients with AML [255], some studies have also suggested that miR-223 can be used as a discriminatory biomarker to distinguish ALL from AML [253,254,285,286,287]. Decreased miR-223 expression in CD19^+^ purified B-cells correlates with very poor prognosis for patients with CLL who are progressing from Binet stage A to C, and can be used to predict their treatment-free survival and overall survival [288,289].

Zhou and colleagues showed that, in Chinese patients with CLL, low miR-223 in CLL cells is associated with increasing cancer stage and worse prognosis [290]. Low miR-223 levels are also associated with the presence of the unmutated immunoglobulin variable heavy chain (IGH) gene in leukemic cells. Unmutated IGH in CLL cells indicates a worse prognosis and aggressive CLL [291]. Moreover, low miR-223 levels are associated with high serum β_2_-microglobulin, which represents a high-risk biomarker for solid tumors, especially colorectal cancers, and is associated with higher mortality in lung and hematological cancer patients [292]. Further, Zhou and colleagues showed that miR-223 levels are low in B-cells isolated from patients with other lymphoproliferative diseases, such as mantle cell lymphoma (MCL) and splenic marginal zone lymphoma (SMZL) [290].

During targeted sequencing of genes relevant for CLL, Rodríguez-Vicente and colleagues identified a common polymorphism (rs2307842) within the 3′UTR of the *HSP90B* gene, encoding heat shock protein 90 alpha family class B member 1 (HSP90AB1), and demonstrated that, in CLL cells, HSP90AB1 is a direct target of miR-223. Low miR-223 and high HSP90AB1 levels are found in clonal B lymphocytes of patients with unmutated IGH CLL and correlate with a shorter time to treatment and a poorer prognosis [293].

High expression of miR-223 and low expression of MOZ are associated with a poor prognosis in AML, particularly in AMoL [272].

Interestingly, miR-223 may regulate cancer immunity. E2A^+^ gastric MALT lymphomas contain abnormal germline B-cells, which express E2A and high levels of miR-223 [294,295]. These lymphomas also show low plasmacytoid dendritic cell (pDC) infiltration. pDCs are effector cells of cancer immunity, and may cause either cancer regression or progression [296,297]. pDC infiltration determines cancer immunity and prognosis in TNBC [297].

Agatheeswaran and colleagues highlighted a negative correlation between miR-223 expression in the peripheral blood of patients with CML and the Sokal score (pre-treatment prognosis predictor at the time of diagnosis), suggesting miR-223 as a disease-risk predictive biomarker in CML [270]. Together, the results discussed here and shown in Table 2 suggest the utility of miR-223 as a biomarker in hematological malignancies.
ijms-25-08191-t001_Table 1Table 1Oncogenic and oncosuppressive functions of miR-223 across the described cancer types, including its impact on cancer treatment response.OrganCancer TypeFunctionDirect Targets and Their Dysregulation in CancerDownstream Molecules ImplicatedSignaling Pathway AffectedCell Process AffectedSponging Activity vs. miR-223Ref.**Intestine**Colorectal carcinomaOncosuppressionFOXO1cyt FOXO1 ↓; nuclear FOXO1 ↑cyclin D1/p21/p27 Cell proliferation ↓; Cell proliferation and invasion ↑lncRNA ROR[61,62]

Onco-miRFOXO3 ↓; FBXW7 ↓; RhoB ↓; RASA1 ↓; SLC4A4 ↓; PRDM1BIM ↓FOXO3a/BIM signaling; FBXW7; PRDM1; Notch and Akt/mTOR pathwaysApoptosis ↓ and cell proliferation ↑; EMT ↑; tumor growth in vivo ↑lncRNA DRAIC; circLRCH3[69,70,71,72,73,74,75,76,77]

Chemoresistance to doxorubicinmiR-223 ↑; FBXW7 ↓N/AMiR-223/FBXW7 axisEMT ↑N/A[78]**Lung**(1) Non-small-cell lung cancer (NSCLC)Oncosuppression NLRP3 ↑N/ANLRP3-mediated inflammasomeCell invasion and migration ↑; inflammation; and innate immunitylncRNA SLCO4A1-AS1 [96]
(i) Adenocarcinoma (ADC)Oncosuppression murine PARP1 ↑N/APARP1-miR-223 negative feedback loopTumor burden ↑; cell proliferation ↑; oxidative stress ↓; antioxidant enzymes, especially catalase ↑; apoptosis and autophagy of cancer cells ↓N/A[99,100]
(ii) Squamous cell carcinoma (SCC)Oncosuppression Mutant TP53 ↑; PTN ↑ N/AMutant TP53 represses the transcription of miR-223 feedback loopCell proliferation and metastasis ↑; cell growth ↑; stemness ↑; and angiogenesis ↑lncRNA PITPNA-AS1[95,98]
NSCLCOnco-miRRhoB ↓; EPB41L3 ↓; TGFBR3 ↓N/A
Cell viability ↑; migration ↑; proliferation ↑; and invasion ↑ADAMTS9-AS2[101,102,103]

Chemoresistance to cisplatin and doxorubicinmiR-223 ↑; FBXW7 ↓ (contradictory reports)N/A
Apoptosis ↓N/A[104,105]

Chemoresistance to erlotinibmiR-223 ↑; FBXW7 ↓ in HCC827/ER cells (contradictory reports)N/A
Apoptosis ↓; colony formation potential ↑; EMT ↑N/A[106]

Chemoresistance to cisplatinmiR-223 ↓; HNMT ↑ (contradictory reports)N/AHER2 signaling pathway  ↑CSCs maintenance and antioxidant propertiesN/A[107]

Chemoresistance to erlotinibmiR-223 ↓; IGF1R ↑ in PC9/ER cells and PC9/CD133+ cells (contradictory reports)N/API3K/Akt signaling pathway Apoptosis ↓N/A[108,109]**Breast**Breast CancerOncosuppressionECT2 ↑, PFN2 ↑; NLRP3 ↑; STIM1 ↑; HAX ↑; FABP7 ↑; SCARB1 ↑; HMGCS1 ↑ IL-1β and IL-18, caspase-9, caspase-7, and caspase-3, ABCA1NLRP3 inflammasomeTumor growth in vivo ↑; cell proliferation, invasion and migration ↑; glycolysis ↑; inflammation ↑; apoptosis ↓; EMT ↑; cholesterol biosynthesis ↑; cholesterol efflux ↓; cancer cell stemness ↑circABCB10; circZFR[131,132,133,137,138,141,143]

Onco-miR*Mef2c ↓*; FBXW7 ↓
Notch signaling; Mef2c-β-catenin pathwayCells invasiveness and metastasization ↑N/A[52,150]

Sensitization to intraoperative RTmiR-223 ↑; EGF ↓N/A
EGF-EGFR pathway (autocrine/paracrine stimulation loop)N/A[154]

Resistance to RTmiR-223 ↓; PFN2 ↑N/AGlycolytic pathway (Warburg effect)Cell viability ↑; glucose consumption, lactic acid production, LDH-A activity, and ATP production ↑circABCB10[133]**Ovary**Ovarian cancerOnco-miRSOX11 ↓; LARP4 ↓; FBXW7 ↓N/AcircBNC2/miR-223/LARP4 and circBNC2/miR-223/ FBXW7 axesCell viability, cell cycle progression; migration, invasion and tumor growth ↑circBNC2/hsa_circ_0008732[13,159,161]

Chemoresistance to cisplatinPTEN ↓PI3K/AKTPTEN/PI3K/AKT pathway ↑ (in vitro and in vivo)Apoptosis ↓; cell viability ↑N/A[51]**Prostate**Prostate cancerOncosuppressionEYA3 ↑c-Myc ↑; CDK2/CDK4 ↑; p21 ↓; p27 ↓ c-Myc signaling pathway ↑Cell proliferation ↑; cell migration and invasion ↑; tumor growth in vivo ↑circGNG4[168]

Onco-miRSEPTIN6 ↓N/A
Cell proliferation ↑; apoptosis ↓; cell invasion ↑N/A[169]

Chemosensitivity to docetaxel miR-223 ↓; FOXO3 ↑N/A
Apoptosis (in vitro) ↑; tumor growth (in vivo) ↑N/A[170]

Resistance to RTmiR-223 ↑; FOXO3 ↓Glut1 ↑; HK-2 ↑; LDH-A ↑
Glycolysis ↑; apoptosis ↓N/A[171]**Central Nervous System**Glioma and glioblastomaOncosuppressionNFIA ↑; MSI2 ↑; CTNND1 ↑; EGFR ↑; NLRP3 ↑p-21 ↑; IL-1β, MCP-1, IL-18, IL-8, and caspase1 ↑ Wnt/β-catenin pathway ↑; PI3K/AKT pathway ↑Cell cycle progression ↑; chemoresistant CSCs differentiation ↓; cell proliferation and migration ↑; EMT ↑; inflammation ↑; apoptosis ↓lncSNHG29 and PITPNA-AS-1[186,187,188,189,190]

Onco-miRPAX6 ↓MMP2, MMP9, and VEGFA ↑
Cell viability and invasiveness ↑N/A[192]

Sensitization to RTmiR-223 ↑; ATM ↓N/A
Tumor growth (in vitro and in vivo) ↓N/A[195]

Chemoresistance to temozolomidemiR-223 ↑; PAX6 ↓N/API3K/AKT pathway ↑
N/A[193,194]**Pancreas**PDACOnco-miRPDS5B ↓; FBXW7 ↓; SLC4A4 ↓PTCH2 ↓; HNRNPK ↑; E-cadherin ↓; Vimentin ↑; MMP2, MMP9, and VEGFA ↑IL6/STAT3/CCND1 axis ↑; Sonic Hedgehog ↑Cell growth, migration and invasion ↑; angiogenesis ↑; apoptosis ↓; tumor growth (in vivo) ↑hsa_circ_001587[220,221,223,224,225,226]

Chemoresistance to cisplatinmiR-223 ↑; FOXO3 ↓N/A
Cell proliferation ↑; apoptosis ↓N/A[229]

Chemoresistance to gemcitabinemiR-223 ↑; FBXW7 ↓E-cadherin ↓; N-cadherin, vimentin, Snail, Slug, ZEB1 and ZEB2 ↑Notch signaling pathway ↑Cell proliferation and migration ↑; apoptosis ↓N/A[222]**Blood**(1) Myeloid cancer







(a) AMLOncosuppressionFBXW7 ↑; PRMT4 ↑; E2F1 ↑N/A
Apoptosis ↓; myeloid differentiation of human stem/progenitor cells ↓N/A[258,259,265]

Onco-miRMOZ ↓N/A
monocyte differentiation ↓; cell proliferation ↑; stemness ↑N/A[272]

Resistance to cisplatinmiR-223 ↑; MOZ ↓N/A
Apoptosis ↓N/A[272]
(b) CMLOncosuppressionMEF2C ↑; PTBP2 ↑; FLT3 ↑Bcl-xL ↑; MMP2PI3K/AKT pathway Cell proliferation ↑; abnormal splicing ↑; cell viability ↑; apoptosis ↓; ROS ↑; tumor growth (in vivo)N/A[270,271]
(2) Lymphoid cancer







(a) ALLOncosuppressionSTAT1 ↑BCL2 ↑
Cell proliferation ↑N/A[268]

Onco-miRARRB1 ↓; FBXW7 ↓TAL1 ↑; JAK2 ↑; HES1 ↑; HES2 ↑; PPARA ↓; DNM1 ↓; GRK4 ↓; MYC ↑; MYB ↑; NOTCH1 ↑; CCNE1 ↑Notch signaling pathway ↑Cell proliferation and invasion of T-ALL cells ↑; apoptosis ↓Circ_0000094[273,276,279]

Chemoresistance to γ-secretase inhibitor miR-223 ↑; FBXW7 ↓N/ANotch signaling pathway ↑Cell proliferation ↑N/A[277]

Chemosensitivity to glucocorticoidsmiR-223 ↑N/A
Apoptosis ↑N/A[283]
(b) CLLOncosuppressionSTAT3 ↑N/A

N/A[269]↑ upregulation; ↓ downregulation; ALL: acute lymphocytic leukemia; AML: acute myeloid leukemia; CLL: chronic lymphocytic leukemia; CML: chronic myeloid leukemia; CSC: cancer stem cells; EMT: epithelial-to-mesenchymal transition; N/A: not available.
ijms-25-08191-t002_Table 2Table 2MiR-223 application as a biomarker across the described cancer types.Organ/TissueCancer TypeChange in miR-223 Levels in Cancer PatientsBiomarker IndicationAccuracy as BiomarkerValues (Confidence Interval) ^1^Involved DownstreamMoleculesReferences**Intestine**CRC ^2^↑ serum/plasma^a^ Diagnosis; ^b^ prognosis (poor)^a^ AUC = 0.963, Se = 97.1%, Sp = 96.7% [79];^b^ AUC = 0.593, Se = 33.3%; Sp = 54.6% [80];^a^ AUC = 0.890 (0.833–0.933) [81];^a^ AUC = 0.838 (0.627–1.000) [82];^a^ AUC = 1, Se = 100%; Sp = 100% in 1st validation set; AUCs = 0.632 and 0.680 in 2nd and 3rd validation sets [85]N/A[80,81,82,83,86]

↓ plasma of CRC patients after surgeryFollow-upN/AN/A[81]

↑ tumorPrognosis: shorter overall survival; worse TNM staging; higher probability to develop metastasesHR = 1.374 (0.708–1.98)N/A[79]

↑ stool and plasmaDiagnosisAUC = 0.796 (0.734–0.858, stool), 0,707 (0.646–0.768, plasma) [83]; AUC = 0.939 (0.825–0.988, stool), Se = 76.5%, Sp = 96.4% [84]N/A[84,85]**Lung**NSCLC↑ NSCLC tissuePrognosis (poor survival)N/AN/A[101]

↑ platelets and platelets-derived MVs of NSCLC patientsDiagnosisN/AN/A[102]

↓ miR-223 and ↑ HNMT expression in NSCLC tissuePrognosis (poor)N/AN/A[107]

↑ sputum/serum/plasma of NSCLC patients^a^ Screening/diagnosis; ^b^ prognosis^a^ within a 12-miR-panel AUC = 0.821 (0.792–0.850), distinguishes NSCLC from HC and COPD [110];^b^ higher risk of progression in ADC patients [110]; ^a^ AUC = 0.94 (0.91–0.96), Se = 87%, Sp = 86% [111]; ^a^ within a 3-miR-panel AUC = 0.951 (0.926–0.976), Se = 84.35, Sp = 90.83 [112];^a^ AUC = 0.809 (0.749–0.860), Se = 69.8%, Sp = 84.3% [113]; ^a^ AUC = 0.744 (0.668–0.811), Se = 76.9%, Sp = 80% [114]; ^a^ AUC = 0.808 (0.712–0.884), Se = 74.3%, Sp = 78.2% [115];^a^ within an 11-miR-panel, accuracies ranging from 0.756 to 0.963 in validation set depending on data-mining technique [116]; N/A [117];^a^ AUC = 0.828 (0.763–0.881) for ADC, Se = 76.8%, Sp = 84.4% [118];N/A [120]; ^a^ AUC = 0.90 (0.81–0.99), Se = 82%, Sp = 95% [121];^a^ AUC = 0.693, Se = 82.1%, Sp = 52% [122];N/A[111,112,113,114,115,116,117,118,119,121,122,123]

↓ serum Diagnosis of early-stage NSCLCAUC = 0.79 (0.64–0.95)
[120]
SCC↓ plasma Earlier disease progression in patients treated with NivolumabN/AN/A[124]**Breast**Breast cancer↑ plasma-derived exosomes of IDC patientsProgression from DCIS to IDCN/AN/A[144]**Ovaries**OC↑ relapsed ovarian serous adenocarcinoma tissueRelapse of ovarian cancerN/AN/A[164]

↑ OC tissuePresence of lymph node metastasis, histological tumor grade, and FIGO stageN/ASOX11 ↓; FBXW7 ↓[13,159]

↑ in OC tissue compared to normal tissue; ↑ serum circulating exosomal miR-223Resistance to cisplatin-based therapy; increased probability of recurrenceN/AN/A[51]

↑ in TAM-derived exosomes Associated with shorter progression-free survival (PFS)N/APTEN ↓; PI3K/AKT ↑ (in recipient OC cells)[51]**Prostate**Prostate Cancer↑ urine and PCa cellsAssociated with radioresistanceN/AN/A[171]

↓ serum of PCa and chronic prostatitis patients vs. BPH patientsDiagnosisAUC PCa vs. non-cancer = 0.817, Se = 81%, Sp = 71%; AUC PCa vs. BPH = 0.938, Se= 88%, Sp = 88%; AUC CP vs. BPH = 0.880, Se = 70%, Sp = 92%N/A[173]**CNS**GBM and Glioma↑ glioma and GBM tissueAssociated with longer survivalN/AMAPK signaling [197]
GBM mesenchymal subtype↑ GBM tissueAssociated with longer survival in the mesenchymal subtypeN/AN/A[198]
GBM↑ GBM tissueAssociated with shorter survivalN/ANFKBIZ; PSCD4; BCL3SLC16A3; PLEKHQ1LHFPL2; LSP1; URP2; CTSL1; ISG20; HMOX1; IL17RA; RBMX; FZD7; CCNB1IP1; LITAF[199]
LGG ↑ LGG tissuePrognosis (poor for high levels)In ROC curves for survival prediction:1-year survival: AUC = 0.6383-year survival: AUC = 0.5775-year survival: AUC = 0.578; better accuracy of a 5-miR signature including miR-223N/A[200]

↑ serum of LGG and GBM patients^a^ Diagnosis; ^b^ follow-up^a^ All gliomas vs. HC: AUC = 0.7771 for miR-223 only, AUC = 0.998 within a 4-miR panel; ^b^ Positive correlation with post-operative MRI score in LGGN/A[202]
Diffuse LGG ↑ whole blood of diffuse LGG (grade II) patients vs. HCDiagnosisAUC = 0.827N/A[201]
 GBM↑ whole blood of GBM patients vs. HC Diagnosis AUC = 0.804N/A[203] **Pancreas**PC↑ whole blood of PC patients vs. HC or CPaDiagnosis Within 4-miR panel: AUC = 0.86 (0.82–0.90), Se = 85%, Sp = 64% in training PC vs. controls; AUC = 0.83 (0.76–0.90), Se = 85%, Sp = 45% in validation PC vs. controls + CPa; other data available
[234]
PC and IPMN↑ tissue and plasma of PC vs. HC and ↑ malignantIPMN patients vs. benign IPMN patients; ↓ in post-operative samplesDiagnosis, prognosis ^a^ with plasma, AUC = 0.834, Se = 62%, Sp = 94.1%;AUC = 0.789, Se = 82.4%, Sp = 62.9% in distinguishing malignant IPMN and PIDC;N/A[235]
IPMN↑ IPMN tissue vs. normal pancreatic tissue (FFPE); ↑ in poor vs. good prognosis IPMN and IPMN vs. CPaDiagnosis, prognosis N/AN/A[236]
PDAC ↑ urine stage I (but not stages II-IV) PDAC patients vs. HCDiagnosisAUC = 0.795 (0.586–1.000), Se = 83.3%, Sp = 76.9%N/A[237]**Blood**ALL↓ Bone marrow of relapsed pediatric ALL patients vs. complete remission patientsLower relapse-free survivalN/AE2F1[280]
ALL↓ in bone marrow and blood of T-ALL and B-ALL pediatric patients compared to HC Diagnosis; follow-upN/AN/A[281]
B-ALL↓ plasma B-ALL vs. HCDiagnosisB-ALL vs. HC Se = 89%, Sp = 100%N/A[287]
AML↓ serum; ↑ in post-operative samples; shorter OS and PFS with low levels ^a^ Diagnosis; ^b^ poor prognosis ^a^ AUC = 0.849, Se = 83.2%, Sp = 81.4% [284]^b^ RR = 3.54 (1.47–5.79), univariate analysis [284];N/A[284]
ALL↓ bone marrow in ALL patients vs. HCDiagnosisN/AN/A[253]
ALL/AML↑ bone marrowDiagnosis; differential diagnosis (AML > ALL > HC)N/A [253];AUC miR-223 only (AML + ALL vs. HC) = 0.853; AUC 3-miR panel = 0.994, Se = 95%, Sp = 98.14% [254];Diagnostic OR 2-miR panel (from metanalysis) = 546, 95% (73.38–4041.28) [285];Accuracy (AML + ALL vs. HC) > 95, Se = 96%, Sp = 95% [286]N/A[254,255,285,286]
CLL↓ miR-223/ ↑ HSP90AB1 (associated to polymorphism rs2307842)Poor prognosis; shorter time to treatment17 months (5–28.9) without treatment vs. 104 months of cases without HSP90AB1 overexpressionHSP90AB1[293]
B-CLLhigh levels in B cells from peripheral bloodHigher overall survival, treatment-free survival, progression free survival137.2 months OS in low vs. not reached in high miR-223 (RR 4.9); 24.1 months TFS in low vs. 107.2 in high miR-223 (RR 2.7) [289]40 months OS in low vs. not reached in high miR-223; 13 months PFS in low vs. not reached in high miR-223 [290]N/A[289,290]
CML↓ in plasmaIncreased disease riskN/AN/A[270]^1^ Accuracy values are reported for miR-223 as a single biomarker when available; as an alternative, accuracy of the panel including miR-223 is provided. ^a^ and ^b^: if miR-223 is indicated as biomarker for more than one application (diagnosis, prognosis, follow-up), these letters will be used to distinguish the accuracies provided for each application; ^2^ CRC: colorectal carcinoma; NSCLC: non-small-cell lung carcinoma; SCC: squamocellular carcinoma; DCIS: ductal carcinoma in situ; IDC: infiltrating ductal carcinoma; OC: ovarian cancer; FIGO: International Federation of Gynecology Obstetrics; PCa: prostate cancer; BPH: benign prostatic hyperplasia; CP: chronic prostatitis; CNS: central nervous system; GBM: glioblastoma; LGG: low grade glioma; PC: pancreatic cancer; CPa: chronic pancreatitis; IPMN: intraductal papillary mucinous neoplasm; PIDC: pancreatic invasive ductal carcinoma; FFPE: formalin-fixed paraffin-embedded; ALL: acute lymphocytic leukemia; AML: acute myeloid leukemia; CLL: chronic lymphocytic leukemia; CML: chronic myeloid leukemia; AUC: area under the curve (ROC analysis); Se: sensitivity; Sp: specificity; HC: healthy controls; OS: overall survival; PFS: progression-free survival; TFS: treatment-free survival; HR: hazard ratio; RR: relative risk; OR: odds ratio; N/A: not available. ↑ increased; ↓ decreased.


## 4. Conclusions and Future Perspectives

From the literature reviewed here, it is clear that the biological role of miR-223 in cancer is very complex, and the general portrait is far from being exhaustively clarified. MiR-223 can have either onco-suppressing (Figure 1 and Appendix A, Table 1) or oncogenic (Figure 2 and Appendix A, Table 1) activities and can affect tumor sensitivity/resistance to chemotherapy based on specific molecular contexts (Table 1). Therefore, a major open question remains: is unraveling the prevalent role played by miR-223 in each cancer type a utopian task?

Here, we conducted a comprehensive review to elucidate the mainstream view on miR-223’s function across different cancers. Most specifically, in breast cancer, NSCLC, and GBM, the prevalent role of miR-223 described in the literature is oncosuppressive, while in ALL, CRC, ovarian cancer, and pancreatic cancer, the oncogenic function is mainly reported. In the case of AML, CLL, CML, and prostate cancer, there is less consensus on the prevalent function of this miRNA during tumorigenesis and progression.

Notably, *NLRP3* and *FBXW7* emerge as consistently reported targets of miR-223 across various cancers, where the miR acts as an oncosuppressor or an oncogene, respectively (Table 1; Figure 1, Figure 2 and Appendix A).

Moreover, we highlighted the relevant clinical use of miR-223 as an easy-to-detect diagnostic and prognostic biomarker in several cancers: in 7 (breast cancer, CLL, CRC, gliomas, NSCLC, ovarian cancer, and pancreatic cancer) out of the 11 cancers analyzed in this review, miR-223 levels are consistently increased in biological fluids (mainly serum or plasma) or in EVs isolated from them as compared to the levels of healthy people or those with lower-risk conditions (Table 2 and Appendix A).

Despite our effort in trying to define an agreement on the function of miR-223 in each type of cancer analyzed, given the complex networks of molecular interactions within cancer cells and their crosstalk with stromal cells, we believe that the unambiguous contribution of miR-223 to a specific cancer type or subtype can be inferred only through the dissection of the tumor microenvironment and further application of technical approaches like single-cell RNA sequencing and in situ hybridization/spatial transcriptomics.

In summary, this review describes the diverse targets and the roles of miR-223 across various prevalent cancer types. Additionally, it offers valuable insights and hints for prospective research on miR-223.

## Figures and Tables

**Figure 1 ijms-25-08191-f001:**
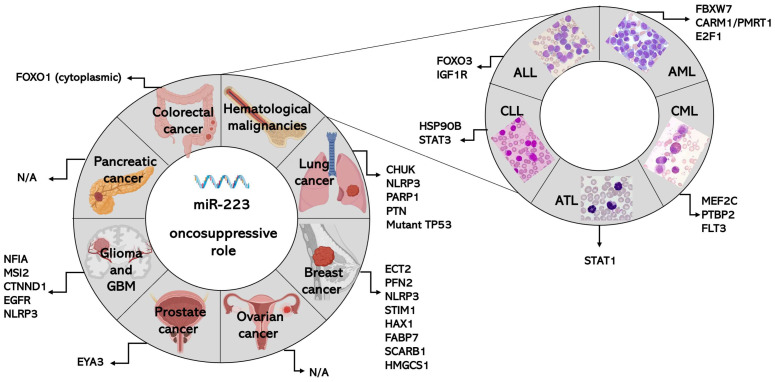
MiR-223 targets implicated in its oncosuppressive function across the depicted tumors. In all cancer models illustrated here, miR-223 has been documented as downregulated, whereas its targets are upregulated (refer to the main text for details) (Abbreviations: GBM: glioblastoma; ALL: acute lymphocytic leukemia; AML: acute myeloid leukemia; ATL: adult T-cell leukemia; CLL: chronic lymphocytic leukemia; CML: chronic myeloid leukemia. N/A: not applicable).

**Figure 2 ijms-25-08191-f002:**
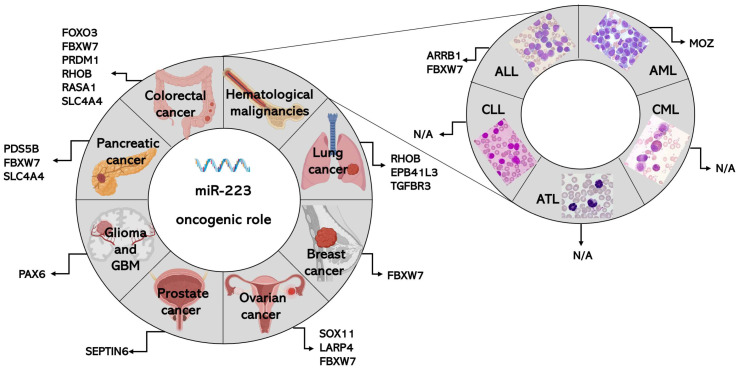
MiR-223 targets implicated in its oncogenic role across the depicted tumors. In all cancer models illustrated here, miR-223 has been described as upregulated, whereas its targets are downregulated (refer to the main text for details) (Abbreviations: GBM: glioblastoma; ALL: acute lymphocytic leukemia; AML: acute myeloid leukemia; ATL: adult T-cell leukemia; CLL: chronic lymphocytic leukemia; CML: chronic myeloid leukemia. N/A: not applicable).

## Data Availability

No new data were created or analyzed in this study. Data sharing is not applicable to this article.

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
