# Peer review of "MiR-223-3p in Cancer Development and Cancer Drug Resistance: Same Coin, Different Faces"

_ijms, 2024, doi:10.3390/ijms25158191_

Round 1
Reviewer 1 Report
Comments and Suggestions for Authors
In this manuscript the authors Barbagallo D. et al have described the oncogenic, oncosuppressive or diagnostic functions of miR-223 in various cancers. The review is detailed and well written. I have a few minor comments:
1) References can be provided for the following sentences:
a) Section 2, line 119: “ MiR-223-3p (henceforth miR-223)…., which maps on chromosome Xq12”.
b) Section 3.3.1, line 474: miR-223 may influence tumor growth…. NLRP3 inflammasome”.
c) Section 3.3.3., line 571: “ Even so, Authors suggests…. Lapatinib”.
d) Section 3.7.1, line 851: “ MiR-223 dirrectly targets…. Pa Tu-8988.”
e) Section 3.8.2., line 971, “ A 3’ UTR polymorphism…. Han Chinese population”.
2) Grammatical error in section 3.1.2. line 267: “ MiR-223 expression is also negatively correlates…(RhoB).” In this sentence remove the word “is” after expression.
3) Section 3.1.4., line 297: “They also showed that higher miR-223 and miR-182 … CRC progression”. The authors can include more details like whether the targets of miR-223 and miR-182 are the same or different. It will be interesting to mention whether these two miRNA drive the CRC progression independent of each other or not.
Reviewer 2 Report
Comments and Suggestions for Authors
The article titled " MiR-223-3p in cancer development and cancer drug resistance: same coin, different faces” by Davide Barbagallo et al. provides valuable insights. However, there are several areas that require attention:
1. In the abstract, authors did not give a clear conclusion or summary of the key findings or implications of the review, which may make it difficult for readers to understand the significance of the research.
2. Authors mentioned the potential use of miR-223 as a diagnostic and prognostic biomarker, but they did not provide clear info on the specificity and sensitivity of miR-223 as a biomarker, nor does it confer potential limitations or challenges in using miR-223 as a biomarker.
3. Authors suggested that the role of miR-223 in cancer is context-dependent and may differ depending on tumor type and subtype, tumor microenvironment, and expression of specific targets, however they not give clear data on how these factors interact or how they can be studied.
4. Authors may provide a schematic representations of dysregulation of miR-223 in the cancers.
5. Authors recommended that a systems and spatial biology approach can be key to understanding the pre-eminent function of miR-223 in a specific tumor, but in the text, they did not provide any info on how such an approach would be implemented.
Comments on the Quality of English LanguageMinor editing of English language required
Reviewer 3 Report
Comments and Suggestions for Authors
The authors provide quite a comprehensive review of the role of miR-223-3p in various cancers and drug resistance. Overall, it summarizes the references cited for the work. Some of the comments are below:
The authors focus quite significantly on cellular studies, cancer tissue microarray analysis, and possibly sequencing. It will help clarify if these are separated with mir expression in various cancers, followed by mechanistic studies. Additionally, what does the normal tissue look like in terms of expressions?
Could they also provide any diagnostic or prognostic values for the mir223? If these are in the table provided, they can be separated. This could be useful since the dual roles are undoubtedly interesting. However, the clinical significance can be emphasized.
Are there mechanistic studies that examined the effect of mir233 knockdown or downregulation in specific cancer types instead of miRNA mimic type overexpression procedures?
Could the author clarify line 348 paragraph? Where they cite Luo. If all the work, such as SK-MES-1 and NCI-H510, are from the same paper, ref 94.?
Could the authors check whether the “m” should be capitalized for microRNAs?
Round 2
Reviewer 2 Report
Comments and Suggestions for Authors
Accept in present form
Comments on the Quality of English LanguageMinor editing of English language required